# A Multi-Expert Ensemble Model for Long-Tailed Steel Surface Defect Detection

## Abstract

In the field of industrial steel surface defect detection, defect images often exhibit a pronounced long-tailed distribution, where tail-class—characterized by scarce samples and subtle features—are much harder to recognize than head classes with abundant data. This imbalance typically results in high miss-detection rates and bias toward head classes. To address this challenge, we propose a Multi-Expert ensemble model that integrates classification and detection experts, introducing a Two-Stage strategy into the classification branch. The framework leverages the complementary strengths of various experts, with validation-based joint optimization of confidence thresholds and expert weights, and employs parallelized training and inference to improve computational efficiency. Experimental results show that the method significantly improves the F1-score (0.912) of tail-class 2 and achieves state-of-the-art average Accuracy (0.989) on the long-tailed Severstal dataset, while strong performance on the balanced NEU dataset further validates its cross-distribution generalizability and practical applicability for industrial steel surface defect detection.

## 1 Introduction

Accurate steel surface defect detection is essential for ensuring product quality and production safety, missing steel defects not only exposes manufacturers to substantial economic losses but also poses serious safety risks. However, ensuring high accuracy in defect detection remains difficult, as defects exhibit significant diversity in their forms and manifestations (Uygun et al., 2024). A common difficulty arises from the long-tailed distribution of existing datasets, as illustrates in Fig. 1, where tail-class 2 with scarce and indistinct samples are particularly hard to detect, thus reducing the overall accuracy. Moreover, with the advancement of industrial manufacturing, real-time detection is increasingly demanded to reduce labor consumption and improve economic efficiency. Conventional detection techniques (Ni et al., 2021; Chu et al., 2017; 2014) are incapable of handling the thousands of images of the steel surface produced in minutes on modern production lines. Therefore, the pursuit of both speed and accuracy in steel defect detection remains a major challenge.

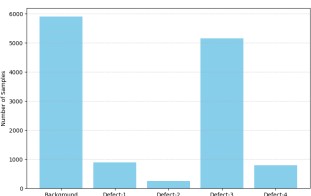

Figure 1: Class distribution of steel surface defects in the Severstal dataset, showing a long-tailed pattern where Defect-2 represents less than 2% of all samples.

Various strategies have been proposed to strengthen tail-class signals for improving recognition accuracy, including data-level augment (Ho et al., 2020; Zhang et al., 2018; Shi et al., 2023), loss-function designs (Cui et al., 2019; Lin et al., 2017), and representation learning techniques (Yan et al., 2024; Kang et al., 2019; Zhu et al., 2022; Du et al., 2024). In the detection of steel surface defects, similar strategies have been widely explored. Firstly, GAN-based synthesis (Liu et al., 2019; He et al., 2019; Yi et al., 2023) has been applied to alleviate data scarcity by generating defect samples, but the complexity of defect textures and morphologies often constrains the realism of the synthesized images. Secondly, Focal Loss integrated into Faster R-CNN (Ren et al., 2015) mitigates class imbalance, but its enhancement of tail classes comes at the cost of reduced head classes accuracy. Then, YOLO (Khanam & Hussain, 2024) and DetectoRS (Qiao et al., 2021) employ multi-scale feature pyramids to handle defects of various scales, can achieve a favorable balance between accuracy and speed. Besides, contrastive learning approaches (Zabin et al., 2023;

Tang et al., 2025) help alleviate the scarcity of labeled samples, but show weakened effectiveness with limited positive–negative sample pairs (Chen et al., 2020). Finally, Transformer-based detectors (e.g., DETR (Zhu et al., 2021), DINO (Zhang et al., 2023)) can achieve high accuracy through global self-attention at the cost of high computational demand, limiting their applicability in real-time industrial scenarios.

In our experiments, we found that head or large-area defects are better addressed by classification models, whereas tail and small-area defects, whose features are difficult to learn, are more effectively handled by detection models for precise localization. To improve overall accuracy, we propose a Multi-Expert ensemble model comprising diverse types of expert with complementary feature-learning abilities, which enhance the recognition of tail and small-area defects. We posit that different experts extract complementary features from the images. Classification experts focus more on global representations such as overall color and shape, whereas detection models, with their localization sensitivity, emphasize textures, edges, and other local patterns, preserving spatial information at the cost of a reduced global receptive field. Therefore, integrating detection and classification experts enables a more comprehensive feature representation and leads to improved recognition performance. In addition, we introduce a Two-Stage strategy into the classification experts to mitigate class imbalance, because we experimented with various tail-class enhancement approaches—such as basic data augmentations(e.g., flipping, rotation, brightness adjustment, etc.), texture augmentation (Riba et al., 2020), and contrastive learning (Oord et al., 2018)—but found the Two-Stage strategy to be the most effective. We further select lightweight experts whose parallel training and inference substantially enhance both detection efficiency and economic benefits. Notably, unlike Mixture-of-Experts(MoE) (Shazeer et al., 2017; Fedus et al., 2022) where experts are jointly trained and selected through a gating mechanism, our approach exploits existing network frameworks, making independent training of each expert more straightforward. The expert weights and confidence thresholds are jointly tuned on the validation set, after which the optimal settings are used for evaluation.

Our proposed method significantly outperforms single-model approaches by dramatically improving tail-class detection while preserving excellent performance on common classes. This effectiveness has been demonstrated across both the long-tailed Severstal dataset and the balanced NEU dataset. The key contributions of our research include:

1. Flexible Multi-Expert Ensemble Framework. We introduce a versatile ensemble architecture that combines classification and detection experts to enhance both tail-class recognition and overall accuracy. The framework's modular design allows seamless integration of diverse expert models and can be adapted to various detection tasks.

2. Computationally Efficient Real-Time Solution. Through parallelized training and inference strategies, our ensemble achieves enhanced efficiency without sacrificing performance quality. This computational optimization makes the system practical for industrial deployment where real-time defect detection is critical.

3. Adaptive Two-Stage Classification Module. We develop an innovative two-stage approach that addresses class imbalance by first distinguishing defective from defect-free samples, then categorizing specific defect types. This hierarchical strategy is model-agnostic and can be easily incorporated into different classification architectures.

4. Robust Cross-Dataset Validation. Extensive experimental evaluation across datasets with contrasting characteristics—the long-tailed Severstal dataset and the balanced NEU dataset—confirms our method's strong generalization capabilities and adaptability to diverse data distributions.

## 2 RELATED WORKS

**Classification Model.** Steel surface defect detection has evolved from traditional image processing to deep learning paradigms. Early methods, such as thresholding (Nand et al., 2014), edge detection (Borselli et al., 2010; Shi et al., 2016), clustering (Melnyk & Tushnytskyy, 2020; Li et al., 2020), and SVMs (Xue-Wu et al., 2011; Agarwal et al., 2011), relied on handcrafted features, which are often inadequate for capturing the complex textures and diverse morphologies of defects. With the emergence of deep learning, convolutional neural networks (CNNs) have demonstrated superior representation capabilities, and pre-trained backbones (e.g., VGGNet (Simonyan & Zisserman,

2014), ResNet (He et al., 2016), DenseNet (Huang et al., 2017), EfficientNet (Tan & Le, 2019), ConvNeXt (Liu et al., 2022)) have been adapted to steel datasets through fine-tuning, significantly improving classification accuracy(Benbarrad et al., 2021; Boudiaf et al., 2022; Guan et al., 2021).

**Detection Model.** Classification models are limited to identifying defect categories, whereas detection models have been developed to further localize defects with precise spatial resolution. For object detection methods, such as Fast R-CNN and Faster R-CNN (Girshick, 2015; Ren et al., 2015) leverage region proposal mechanisms to deliver high-precision detection, while Cascade R-CNN (Cai & Vasconcelos, 2018) and DetectoRS (Qiao et al., 2021) further enhance bounding box localization accuracy through multi-stage IoU thresholds and recursive feature pyramid, thereby improving small-area object localization. Although lightweight variants (Ren et al., 2018; Wang et al., 2021a; Akhyar et al., 2023) improve efficiency for steel industrial applications, their robustness remains limited when confronted with irregular and unstable defect morphologies. Another method, such as the YOLO series (Redmon et al., 2016; Ge et al., 2021; Khanam & Hussain, 2024) and SSD (Liu et al., 2016), are characterized by their end-to-end architecture and real-time detection capability. Recently, their performance on small-area objects and complex backgrounds has been progressively enhanced through multi-scale feature fusion and anchor-free designs, and numerous studies (Li et al., 2022; Min et al., 2022; Zhai et al., 2020; Liu & Gao, 2021) have confirmed the strong capability of such methods.

**Transformer-based Model.** Transformer-based detectors such as DETR (Zhu et al., 2021) and DINO (Zhang et al., 2023) have recently achieved high accuracy in complex, multi-scale defect scenarios. These methods typically leverage self-attention mechanisms to model global dependencies, thereby enabling the capture of long-range features. However, their large model size and computational overhead (Liu et al., 2024; Cao et al., 2025; Zhang et al., 2025) limit their practicality in industrial applications.

**Model Ensemble.** Early studies on defect detection primarily relied on single-model architectures, which have gradually reached performance bottlenecks. To address this problem, the MoE method has been proposed, where multiple specialized expert networks are dynamically activated or combined to harness their complementary strengths (Shazeer et al., 2017). While MoE has achieved remarkable success in large language models (Zadouri et al., 2024; Yan et al., 2025), its application to industrial defect detection remains largely unexplored. Unlike ensemble methods (Ganaie et al., 2022), which typically fuse outputs from multiple classifiers in a heuristic or weighted manner, and has been widely explored to defect detection(Chen et al., 2018; Konovalenko et al., 2022; Wang et al., 2021b), but they often rely on simple fusion of a few classifier outputs.

## 3 METHODOLOGY

In this section, we introduce an expert ensemble framework for steel defect detection that leverages weighted combination of multiple specialized models. The methodology employs a Two-Stage classification strategy: an initial binary classifier separates defective samples from defect-free ones, followed by a multi-class classifier that categorizes specific defect types. Performance optimization is achieved through iterative joint tuning of confidence thresholds $\{\delta_k\}$ and ensemble weights $\{w_k\}$. The resulting aggregated predictions demonstrate enhanced recognition performance for underrepresented defect classes while preserving competitive accuracy on common defect types. Complete algorithmic specifications and ensemble implementation details are documented in Appendix A.

### 3.1 MULTI-EXPERT ENSEMBLE MODEL

Our multi-expert ensemble framework combines both classification and detection models (Fig. 2), where each expert produces binary predictions based on learned confidence thresholds $\delta_k$. The final prediction emerges from a weighted combination of expert outputs, with weights $w_k$ optimized to leverage each model's unique strengths.

We begin by independently training each expert using its specialized loss function, creating a comprehensive model pool $\mathcal{E} = \{E_1, \ldots, E_m\}$. For computational efficiency, we select a subset $\mathcal{E}_{\text{sel}} \subseteq \mathcal{E}$ containing only the $K$ fastest-inference experts, forming our active ensemble of size

$|\mathcal{E}_{\text{sel}}| = K$. The ensemble learning objective combines individual expert losses as:

$$\mathcal{L}_{\text{Exp}} = \sum_{E_k \in \mathcal{E}_{\text{sel}}} w_k \, \mathcal{L}_{E_k}.$$

Rather than using simple averaging or majority voting, we adaptively optimize the expert weight $\{w_k\}$ through validation-based tuning. This optimization problem seeks the best threshold and weight configuration:

$$\{\delta^*, w^*\} = \arg\max_{\delta,\, w} \, \text{ValMetric}(\delta, w) \quad \text{s.t.} \sum_{k=1}^{K} w_k = K, \; w_k \geq 0, \; 0 < \delta_k < 1.$$

The validation metric $\text{ValMetric}(\delta, w)$ is defined as the sum of Precision and Recall scores. We solve this optimization thorough an alternating approach: first fixing weights $\{w_k\}$ to find optimize thresholds $\{\delta_k\}$, then fixing thresholds $\{\delta_k\}$ to optimize weights $\{w_k\}$. This iterative process typically converges within 2-3 cycles, making it computationally practical for real-world deployment.

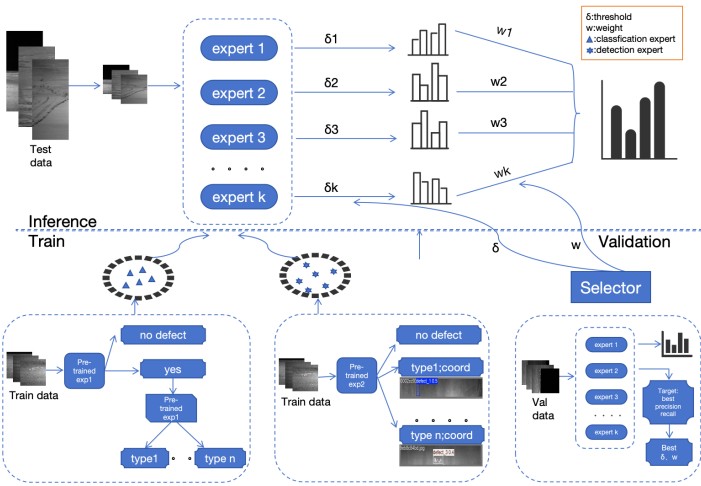

Figure 2: Schematic of the proposed a multi-expert ensemble model, integrating independently trained classification and detection experts with validation-based threshold and weight optimization, and employing a Two-Stage classification strategy to alleviate class imbalance.

To demonstrate our approach, we incorporate popular detection models like Faster R-CNN (Ren et al., 2015) and YOLO (Redmon et al., 2016), alongside classification experts such as ResNet (He et al., 2016) and VGG (Simonyan & Zisserman, 2014). For classification experts, we employ Binary Cross-Entropy with Logits Loss (Shannon, 1948), which excels at multi-label tasks by treating each category as an independent binary decision:

$$\mathcal{L}_{E_1} = -\frac{1}{S \cdot C} \sum_{s=1}^{S} \sum_{c=1}^{C} \left[ y_{s,c} \log \sigma(x_{s,c}) + (1 - y_{s,c}) \log \left( 1 - \sigma(x_{s,c}) \right) \right],$$

where $S$ denotes the number of samples, and $C$ denotes the total number of defect classes.

Detection experts use more complex loss formulations. Faster R-CNN (Ren et al., 2015) optimizes both classification and regression at two stages: the Region Proposal Network (RPN) and Region of Interest (RoI) phases:

$$\mathcal{L}_{E_2} = \mathcal{L}_{\text{RPN}}^{\text{cls}} + \mathcal{L}_{\text{RPN}}^{\text{reg}} + \mathcal{L}_{\text{RoI}}^{\text{cls}} + \mathcal{L}_{\text{RoI}}^{\text{reg}}$$

$$= \sum_{i \in \text{RPN}} \left[ -\alpha_t (1 - p_t^{(i)})^\gamma \log(p_t^{(i)}) \right] + \sum_{i \in \text{RPN}} \sum_{j=1}^{J} R\left( \left| t_j^{(i)} - \hat{t}_j^{(i)} \right| \right)$$

$$+ \sum_{k \in \text{RoI}} \left[ -\sum_{s=1}^{S} y_s^{(k)} \log(p_s^{(k)}) \right] + \sum_{k \in \text{RoI}} \sum_{j=1}^{J} R\left( \left| t_j^{(k)} - \hat{t}_j^{(k)} \right| \right).$$

The RPN classification loss uses Focal Loss (Lin et al., 2017) to address foreground–background imbalance, with $p_t^{(i)} = p^{(i)}$ if $y^{(i)} = 1$ and $1 - p^{(i)}$ otherwise, parameters $\alpha_t$ for class balancing, and $\gamma$ for the focusing parameter. The regression components $R(\cdot)$ employ L1 loss to predict accurate bounding box coordinates, with $t_j$ and $\hat{t}_j$ represent predicted and target box parameters, $y_s^{(k)}$ and $p_s^{(k)}$ denote the ground-truth one-hot label and predicted class probability of proposal $k$.

Similarly, YOLO-based experts (Redmon et al., 2016) optimize a balanced combination of three objectives:

$$\mathcal{L}_{E_3} = \lambda_{\text{box}} \cdot \mathcal{L}_{\text{CIoU}} + \lambda_{\text{cls}} \cdot \mathcal{L}_{\text{BCE-cls}} + \lambda_{\text{obj}} \cdot \mathcal{L}_{\text{BCE-obj}}.$$

This formulation includes Complete Intersection over Union (CIoU) loss, $\mathcal{L}_{\text{CIoU}}$, for precise localization, multi-class cross-entropy ($\mathcal{L}_{\text{BCE-cls}}$) for category prediction, and binary cross-entropy for object detection confidence ($\mathcal{L}_{\text{BCE-obj}}$). The weighting parameters $\lambda_{\text{box}}, \lambda_{\text{cls}}, \lambda_{\text{obj}}$ control the relative importance of each component.

Once individual training completes, we combine expert predictions through our adaptive weighting ensemble scheme. Both decision confidence thresholds and expert importance weights undergo iterative optimization using validation data, ensuring the selected expert subset $\mathcal{E}_{\text{sel}}$ achieves optimal performance for the specific defect detection task.

### 3.2 Two-Stage Strategy In Classification Experts

To mitigate class imbalance and enhance tail-class performance, we adopt a Two-Stage classification strategy: first, a binary classifier separates defective samples from defect-free ones, then a specialized multi-label classifier categorizes the specific defect types within the identified defective samples. This hierarchical design offers several advantages: it reduces computational costs by early filtering of clean samples, minimizes false positive detections, and maintains flexibility by allowing different classification architectures to be easily integrated. Therefore, $\mathcal{L}_{E_1}$ is reformulated as:

$$\mathcal{L}_{E_1} = \mathcal{L}_{\text{binary}} + \mathcal{L}_{\text{multilabel}} = -\frac{1}{S} \sum_{s=1}^{S} \text{CE}(y_s^{\text{bin}}, p_s^{\text{bin}}) - \frac{\lambda}{|\mathcal{S}_{\text{def}}|} \sum_{s \in \mathcal{S}_{\text{def}}} \sum_{c=1}^{C} \text{CE}(y_{s,c}, p_{s,c}),$$

where the binary stage loss handles the defect/no-defect decision and the multi-label stage loss focused only on samples identified as defective. Here $\mathcal{S}_{\text{def}}$ represents the subset of defective samples, $y_s^{\text{bin}}$ and $p_s^{\text{bin}}$ are the ground truth and predicted binary labels for defect presence, while $y_{s,c}$ and $p_{s,c}$ denote the multi-label annotations and predictions for specific defect categories. The $\lambda$ is a hyperparameter, and CE represents the standard cross-entropy loss function.

## 4 Datasets

Our experiments utilize two distinct datasets: the Severstal dataset and the NEU dataset. The Severstal dataset presents pronounced long-tailed distribution and complex defect textures and morphologies. In contrast, the NEU dataset exhibits a more balanced class distribution. Given that the Severstal dataset better reflects the characteristics of real-world industrial defect detection scenarios, we use it as our main evaluation benchmark throughout the experimental analysis.

### 4.1 Severstal Dataset

The Severstal steel defect dataset (Kaggle, 2019) comprises 12,568 high-resolution steel images ($1600 \times 256$ pixels) with detailed pixel-level defect annotations. The dataset categorizes defects into 4 types: (1) Pitted Surface, (2) Crazing, (3) Scratches, and (4) Patches. The class distribution exhibits several imbalance, with Scratches (Class 3) accounting for approximately $40\%$ of all defective samples, whereas Crazing (Class 2) constitutes less than $2\%$ of cases, leading to a strongly long-tailed distribution pattern (see detailed statistics in Appendix Table 3). We split the data using an 8:1:1 ratio for training, validation, and testing, with all hyperparameter optimization performed on the validation subset.

The four defect types display distinct visual characteristics (illustrated in Fig. 3). Pitted Surface defects (Class 1) appear as small, dispersed regions across the steel surface. Crazing (Class 2) manifests as narrow vertical scratch patterns with consistent morphology but occurs very infrequently.

Scratches (Class 3) present as extensive abrasions that cover large surface areas. Patches (Class 4) consist of irregular raised or depressed regions with significant variation in shape and appearance within the class.

This dataset's challenging characteristics, including extreme class imbalance and diverse defect morphologies, make it an ideal testbed for evaluating robust defect detection systems in realistic industrial settings.

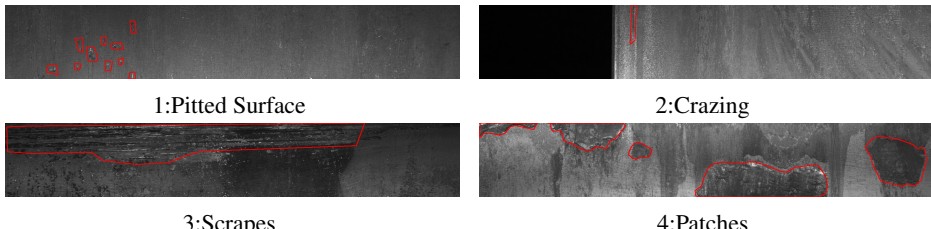

1:Pitted Surface          2:Crazing

3:Scrapes          4:Patches

Figure 3: Representative examples of the four defect categories in the Severstal dataset, highlighting their distinct morphological characteristics.

### 4.2 NORTHEASTERN UNIVERSITY SURFACE DEFECT DATABASE

The Northeastern University (NEU) surface defect dataset, introduced by Song & Yan (2013), comprises 1,800 grayscale images of hot-rolled steel strips(resolution $200 \times 200$). It includes six balanced categories (300 samples each): Crazing (Cr), Inclusion (In), Patches (Pa), Pitted Surface (Ps), Rolled-in Scale (Rs), and Scratches (Sc), as illustrated in Fig. 4. Owing to its multiple versions , the NEU dataset has been widely adopted for steel surface defect classification, detection, and segmentation tasks (Zhao et al., 2021; Sharma et al., 2022). In this study, we adopt the standard 8:1:1 split for training, validation, and testing, ensuring uniform class distribution across subsets.

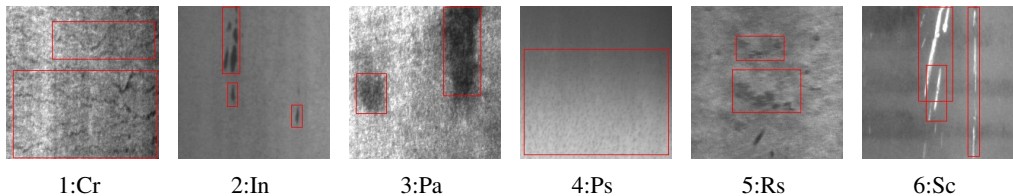

1:Cr      2:In      3:Pa      4:Ps      5:Rs      6:Sc

Figure 4: Representative examples of the six defect categories in the NEU dataset.

## 5 EXPERIMENT

We evaluate the proposed approach on the Severstal and NEU datasets using representative detection models and convolutional neural network(CNN) classifiers. For the long-tailed Severstal dataset, the proposed Two-Stage strategy effectively improves performance of tail-class; whereas on the balanced NEU dataset, competitive performance can be achieved with One-Stage classification models alone. In addition, we discuss the stability of the proposed method in Appendix E. Due to page limitations, we provide a detailed report of the experimental results on the NEU dataset in Appendix G.

### 5.1 EXPERIMENTAL SETUP

All experiments were conducted using PyTorch framework on a server equipped with an RTX 4090 GPU. We evaluated four different classification architectures: EfficientNet-B3, ConvNeXt-Small, ResNet101, and VGG19. These models were trained on images resized to $224 \times 224$ pixels, utilizing Binary Cross-Entropy with Logits loss and Stochastic Gradient Descent optimization with 0.9 momentum. Training proceeded for a maximum of 60 epochs with early stopping mechanisms implemented to prevent overfitting.

For detection tasks, we tested six models: Faster R-CNN, Cascade R-CNN, DetectoRS, DINO, YOLO8, and YOLO11. The first three models (Faster R-CNN, Cascade R-CNN, and DetectoRS) underwent 24-epoch training cycles with conventional data augmentation techniques including CutOut, random horizontal flipping, random cropping, and brightness adjustments, all optimized using SGD with momentum. DINO required extended training of 60 epochs with random flipping augmentation and Adam optimizer. The YOLO series (YOLO8 and YOLO11) were trained for 100 epochs. Four models are trained at full resolution ($1600 \times 256$) to preserve the original image detail.

## 5.2 Two-Stage vs. Other Tail-Class Enhancements

Our Two-Stage strategy demonstrates clear advantages over alternative enhancement techniques on the challenging long-tailed Severstal dataset, with comprehensive results presented in Table 1. The best results or our method are highlighted in bold. The same convention applies throughout and will not be repeated. Several competing methods show limited effectiveness: texture-based augmentation (Riba et al., 2020) provides minimal improvements, which we attribute to the irregular and poorly-defined boundaries characteristic of steel defects that reduce the utility of edge-based features. Contrastive learning (Oord et al., 2018) similarly underperforms, likely hampered by the dataset's relatively small sample size that limits the method's ability to learn robust representations.

Standard data augmentation techniques do provide measurable benefits by introducing realistic variations in defect orientation and imaging conditions. However, our Two-Stage framework achieves the most substantial improvements by systematically addressing class confusion and mitigating the effects of severe class imbalance inherent in the dataset. The results consistently show that both "Two-Stage" and "Two-Stage + augment" configurations outperform other methods across all tested classification architectures, demonstrating the robustness of our approach.

Additional analysis in the Appendix (Table 4) provides detailed Accuracy breakdowns for individual defect classes. Most significantly, our Two-Stage strategy is the only method that simultaneously improves the Accuracy ($ACC$) of both the tail-class (class 2) and the head-class (class 3) across all backbones. This dual improvement on both tail and head classes provides strong evidence for the effectiveness and practical value of our hierarchical classification approach.

Table 1: Average $ACC$ across all classes under different tail-class enhancement strategies. "Base" denotes the One-Stage baseline. "Texture" applies texture-based augmentation. "Contrastive" indicates contrastive learning. "Augment" includes standard augmentations such as rotation, cropping, and brightness adjustment. "Two-Stage" is our proposed strategy, and "Two-Stage+Augment" combines it with augmentation. $\Delta\%$ represent the relative improvement over the Base method.

| Method | Base | Texture | Contrastive | Augment | **Two-Stage** | **Two-Stage+Augment** |
|---|---|---|---|---|---|---|
| Efficient b3 | 0.9609 | -0.92% | -0.35% | -0.14% | **+0.23%** | **+0.85%** |
| ConvNeXt small | 0.9695 | -0.48% | -0.00% | +0.48% | **+0.14%** | **+0.78%** |
| ResNet101 | 0.9599 | +0.50% | +0.25% | +0.42% | **+0.77%** | **+1.28%** |
| VGG19 | 0.9723 | -0.42% | -0.26% | -0.05% | **+0.02%** | **+0.07%** |

## 5.3 Ablation Study

We conducted an ablation study on the Severstal dataset to investigate how ensemble size affects performance, systematically testing configurations with 2 through 5 expert models. The results, visualized in Fig. 5, reveal clear trends regarding ensemble composition.

Firstly, the boxplot analysis demonstrates that adding more experts consistently enhances overall performance while simultaneously reducing prediction variance across evaluation metrics. This suggests that larger ensembles not only achieve better average results but also provide more stable and reliable predictions. Secondly, the radar chart comparison of top-performing ensembles shows that the five-expert configuration delivers the most well-rounded performance profile, with particularly strong improvements in Average Precision and Overall Accuracy metrics. It is important to note that Overall Accuracy denotes the fraction of samples for which every class is correctly classified, making it a stringer measure of system-wide performance.

Based on these findings, we adopt the largest computationally feasible ensemble size for all subsequent experiments. This decision ensures we maximize both the robustness of our predictions and the consistency of performance across different defect types and imaging conditions, which is crucial for reliable industrial deployment. However, it is crucial to understand that the improved performance with increasing ensemble size depends fundamentally on how well the individual experts complement each other, not merely on having more models in the ensemble.

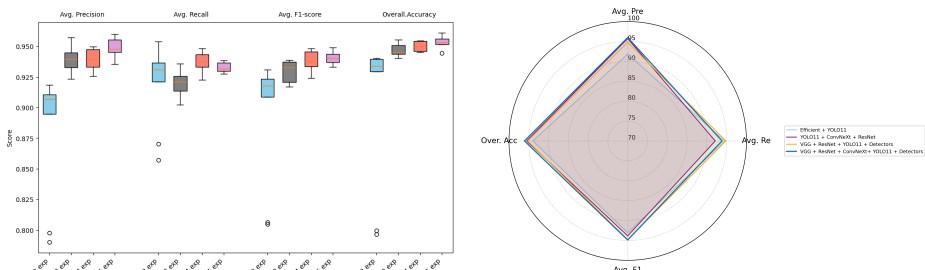

(a) Boxplots of model performance with varying numbers of experts

(b) Radar plots of the best-performing ensembles based on F1-score

Figure 5: Performance comparison of ensembles with different numbers of experts on the Severstal dataset. Panel (a) corresponds to Table 5–Table 8 in the appendix, while panel (b) presents the best-performing combinations (F1-score). All classification experts employed the Two-Stage strategy.

## 5.4 Experimental Results on the Severstal Dataset

To enhance computational efficiency, we excluded three resource-intensive models (DetectoRS, DINO, Cascade R-CNN), and combined the remaining seven experts into our ensemble. Since our primary focus is accurate defect classification rather than precise spatial localization, we extracted only the class confidence scores from detection models like YOLO while discarding their bounding-box predictions.

Performance results on the Severstal test set are presentd in Table 2 for F1-score and Accuracy, with additional Precision and Recall metrics detailed in the Appendix Table 9–Table 10. The analysis reveals distinct strengths and weaknesses among different model types: Classification models generally underperform on the tail and small-area defect (Class 2), whereas detection models such as Faster R-CNN, Cascade R-CNN, and DetectoRS show limited robustness on Class 4 with irregular shapes. Our ensemble approach addresses these individual model limitations effectively. The best configuration combines all seven selected experts with our Two-Stage strategy, achieving the best overall performance. This result validates that strategic ensemble combination can overcome the specific weaknesses of individual architectures while amplifying their complementary strengths.

An interesting observation is that, while the Two-Stage+Augment strategy yields superior performance for individual experts, its advantage diminishes within the ensemble setting. The diminished ensemble benefit highlights an importance trade-off: while standard augmentation helps individual models become more robust, it can inadvertently reduce the complementary nature of different experts. Since ensemble effectiveness relies heavily on combining diverse perspectives, the loss of inter-model diversity ultimately limits the collective performance gains. Thus, ensemble design requires careful consideration of how preprocessing and training strategies affect model diversity, and that techniques beneficial for individual models may not necessarily translate into improved ensemble performance.

The proposed multi-expert ensemble method establishes new performance benchmarks on the dataset, achieving an exceptional average F1-score of $0.9498$ and average Accuracy of $0.9892$. Most significantly, it markedly improves detection of the tail-class 2 ($F1_2 = 0.9123$, $ACC_2 = 0.9960$), demonstrating that our method can substantially enhance recognition of tail-class defects. These results validate two critical aspects of our framework: first, it achieves superior overall performance compared to state-of-the-art methods, and second, it specifically addresses the long-standing challenge of reliably detecting rare defect types that are often missed by conventional approaches. The

combination of excellent overall metrics with strong tail-class performance indicates that our ensemble strategy successfully balances comprehensive defect coverage with high-precision.

Specifically, the results on the NEU dataset are provided in Appendix G. The experiments demonstrate that our method performs well not only on long-tailed datasets but also achieves high performance on balanced datasets.

Table 2: Per-class F1-score ($F1_i$) and Accuracy ($ACC_i$), together with their averages ($F1_{avg}$, $ACC_{avg}$), on the Severstal test set. Here, $F1_i$ (i=0–4) denote the per-class F1-score for background and four defect categories. Methodological variants:(1s)One-Stage strategy in the classification branch; (2s)Two-Stage strategy;(2s_aug)Two-Stage strategy with data augmentation techniques. Voting_7 denotes the configuration in which all model weights are set to 1.

| Expert | $F1_0$ | $F1_1$ | $F1_2$ | $F1_3$ | $F1_4$ | $F1_{avg}$ | $ACC_0$ | $ACC_1$ | $ACC_2$ | $ACC_3$ | $ACC_4$ | $ACC_{avg}$ |
|---|---|---|---|---|---|---|---|---|---|---|---|---|
| Efficient_1s | 0.9392 | 0.7952 | 0.7097 | 0.8905 | 0.9171 | 0.8503 | 0.9412 | 0.9729 | 0.9857 | 0.9165 | 0.9880 | 0.9609 |
| ConvNeXt_1s | 0.9511 | 0.8831 | 0.8070 | 0.9056 | 0.9622 | 0.9018 | 0.9539 | 0.9857 | 0.9913 | 0.9220 | 0.9944 | 0.9695 |
| ResNet101_1s | 0.9356 | 0.8024 | 0.7059 | 0.8868 | 0.9140 | 0.8489 | 0.9371 | 0.9737 | 0.9880 | 0.9133 | 0.9872 | 0.9599 |
| VGG19_1s | 0.9602 | 0.8421 | 0.7857 | 0.9183 | 0.9556 | 0.8924 | 0.9618 | 0.9809 | 0.9905 | 0.9347 | 0.9936 | 0.9723 |
| Efficient_2s | 0.9408 | 0.8000 | 0.7778 | 0.8995 | 0.8671 | 0.8570 | 0.9419 | 0.9769 | 0.9904 | 0.9244 | 0.9817 | 0.9631 |
| ConvNeXt_2s | 0.9514 | 0.8375 | 0.8519 | 0.9259 | 0.9091 | 0.8952 | 0.9539 | 0.9793 | 0.9936 | 0.9411 | 0.9865 | 0.9709 |
| ResNet101_2s | 0.9471 | 0.7945 | 0.7451 | 0.9133 | 0.9341 | 0.8668 | 0.9490 | 0.9761 | 0.9896 | 0.9316 | 0.9904 | 0.9673 |
| VGG19_2s | 0.9612 | 0.8228 | 0.8000 | 0.9249 | 0.9282 | 0.8874 | 0.9626 | 0.9777 | 0.9912 | 0.9412 | 0.9896 | 0.9725 |
| Efficient_2s_aug | 0.9542 | 0.7692 | 0.7941 | 0.9197 | 0.9457 | 0.8766 | 0.9570 | 0.9714 | 0.9889 | 0.9364 | 0.9920 | 0.9691 |
| ConvNeXt_2s_aug | 0.9593 | 0.8931 | 0.7778 | 0.9435 | 0.9457 | 0.9039 | 0.9618 | 0.9865 | 0.9905 | 0.9547 | 0.9920 | 0.9771 |
| ResNet101_2s_aug | 0.9536 | 0.8421 | 0.7111 | 0.9244 | 0.9497 | 0.8762 | 0.9554 | 0.9809 | 0.9897 | 0.9419 | 0.9928 | 0.9722 |
| VGG19_2s_aug | 0.9591 | 0.8121 | 0.8364 | 0.9347 | 0.9153 | 0.8915 | 0.9610 | 0.9753 | 0.9928 | 0.9475 | 0.9881 | 0.9730 |
| Faster R-CNN | 0.6324 | 0.7624 | 0.8679 | 0.7780 | 0.5946 | 0.7271 | 0.7382 | 0.9658 | 0.9944 | 0.7780 | 0.9045 | 0.8762 |
| Cascade R-CNN | 0.4839 | 0.7380 | 0.7333 | 0.7216 | 0.5825 | 0.6519 | 0.6690 | 0.9610 | 0.9873 | 0.7016 | 0.8974 | 0.8553 |
| DetectoRS | 0.7864 | 0.8608 | 0.8519 | 0.8683 | 0.6380 | 0.8011 | 0.8250 | 0.9825 | 0.9936 | 0.8870 | 0.9196 | 0.9215 |
| DINO | 0.9653 | **0.9114** | 0.8462 | 0.9331 | **0.9787** | 0.9269 | 0.9666 | **0.9889** | 0.9936 | 0.9475 | 0.9968 | 0.9787 |
| YOLO8 | 0.9578 | 0.8790 | 0.8462 | 0.9340 | 0.9570 | 0.9148 | 0.9602 | 0.9849 | 0.9936 | 0.9459 | 0.9936 | 0.9756 |
| YOLO11 | 0.9678 | 0.8889 | 0.8889 | 0.9340 | 0.9551 | 0.9269 | 0.9690 | 0.9865 | 0.9952 | 0.9475 | 0.9936 | 0.9784 |
| Voting_7 | 0.9677 | 0.9020 | 0.8235 | 0.9414 | 0.9724 | 0.9214 | 0.9690 | 0.9881 | 0.9928 | 0.9531 | 0.9960 | 0.9798 |
| **Ours_1s** | 0.9684 | 0.8961 | 0.8235 | 0.9436 | 0.9724 | 0.9208 | 0.9698 | 0.9873 | 0.9928 | 0.9547 | 0.9960 | 0.9801 |
| **Ours_2s** | **0.9958** | 0.9020 | **0.9123** | **0.9610** | 0.9780 | **0.9498** | **0.9960** | 0.9881 | **0.9960** | **0.9690** | 0.9968 | **0.9892** |
| **Ours_2s_aug** | 0.9901 | 0.8961 | 0.8302 | 0.9594 | 0.9836 | 0.9319 | 0.9905 | 0.9873 | 0.9928 | 0.9674 | **0.9976** | 0.9871 |

## 5.5 TRAINING AND INFERENCE EFFICIENCY

The training and inference efficiency of different methods on the Severstal and NEU datasets is summarized in Fig. 6. Overall, detection models (e.g., DINO, DetectoRS, Cascade R-CNN) exhibit substantially higher training and inference latency, whereas lightweight classifiers such as EfficientNet-B3 and ConvNeXt-Small achieve the fastest performance. The proposed ensemble framework benefits from parallelization, with inference time primarily constrained by the slower Faster R-CNN expert, yet takes significantly less time than the most computationally expensive detection models.

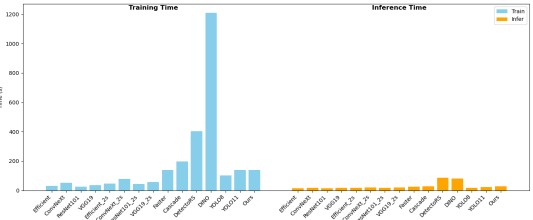

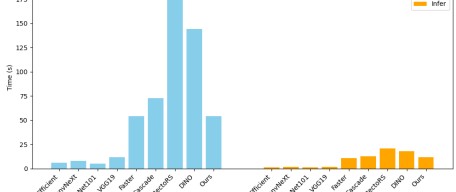

(a) Computational efficiency on the Severstal dataset.

(b) Computational efficiency on the NEU dataset.

Figure 6: Comparison of training (per-epoch) and inference efficiency across different methods on the left Severstal dataset and the right NEU dataset.

## 6 CONCLUSION

This research addresses the critical problem of detecting rare classes in imbalanced datasets by developing a multi-expert ensemble framework that combines object detection models with CNN classifiers with a Two-Stage classification strategy. Experimental results on the long-tailed Severstal dataset show enhanced detection recognition across most categories, with particularly significantly improvements in capturing previously missed tail-class defects. The method also demonstrates robust generalization on the balanced NEU dataset, achieving top-tier or near-optimal results across all evaluation criteria. An important practical benefit of our appraoch is its computational efficiency achieved through parallel processing, making it suitable for deployment in industrial environments requiring real-time defect detection capabilities.

## REPRODUCIBILITY STATEMENT

All datasets used in our experiments are publicly available. Specifically, the Severstal dataset and the NEU surface defect dataset can be directly accessed through their official sources. Furthermore, the implementation of our proposed multi-expert ensemble model, including training scripts and evaluation protocols, will be made publicly available on GitHub upon publication. Moreover, all experiments were performed with fixed random seeds, thereby guaranteeing the complete reproducibility of our results.

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

# A ALGORITHM

The proposed approach is summarized in Algorithm 1, where the **decision rules** for ensembling predictions from multiple experts are defined as follows.

(1) If at least $K - 1$ experts (where $K$ is the total number of experts) vote for the background (non-defect) class, the sample is directly classified as background and no further defect inference is performed.

(2) For each defect category, if the number of supporting votes exceeds half of the experts, that defect is regarded as present.

(3) If no defect category receives votes surpassing half of the experts, the prediction defaults to background (non-defect).

Together, these rules establish a robust consensus mechanism, not only maintain high precision in recognizing head classes but also enhance the detection of tail defects, thereby enhancing the overall reliability of the proposed multi-expert ensemble model.

# B DETAILED DESCRIPTION OF THE SEVERSTAL DATASET

In the Severstal dataset, the labeled set comprises 12,568 images, including 5,902 background samples (46.96%), where each image may contain multiple defect categories simultaneously..Among the defect categories, Class 3 accounts for approximately 40% of the samples, whereas Classes 1, 2, and 4 each represent less than 10%, exhibiting a pronounced long-tailed distribution, as shown in Table 3. The dataset is split into train, validation, and test sets with a ratio of 8:1:1, where hyperparameter tuning is performed on the validation set, and the final performance is evaluated on the test set.

---

**Algorithm 1:** Multi-Expert Ensemble: Training + Validation Tuning + Inference

---

**Input:** Train set $\mathcal{D}_{\text{train}}$, validation set $\mathcal{D}_{\text{val}}$, test set $\mathcal{D}_{\text{test}}$, classification expert set $\mathcal{C}$ (Two-Stage), detection expert set $\mathcal{D}$, early stopping patience patience, maximum epochs $E_{\max}$, threshold grid $\Delta$, weight constraint set $\mathcal{W}$

**Output:** Optimal thresholds $\{\delta_k^\star\}$, optimal weights $\{w_k^\star\}$, test predictions $y_{\text{test}}$

1 **Phase A: Independent Expert Train on the $\mathcal{D}_{\textbf{train}}$**;

2 **foreach** $M_i^{cla} \in \mathcal{C}$ **do**

3     Train Stage-1 (binary classification) and Stage-2 (multi-label classification) until early stopping

4 **foreach** $M_j^{det} \in \mathcal{D}$ **do**

5     Train with standard detection losses until early stopping

6 **Phase B: Validation-based Alternating Parameter Tuning**;

7 Initialize weights $w_k = 1, \sum_{k=1}^{K} w_k = K; \delta_k \in (0,1)$;

8 **repeat**

9     Fix $\{w_k\}$ and search over $\Delta$ for the optimal thresholds $\{\delta_k\}$ maximizing the validation metric;;

10     Fix $\{\delta_k\}$ and search within $\mathcal{W}$ for the optimal weights $\{w_k\}$;

11 **until** *convergence or maximum iterations*;

12 Denote $\delta_k^\star, w_k^\star$ as the final tuned parameters;

13 **Phase C: Test Set Inference**;

14 **foreach** $(I, \cdot) \in \mathcal{D}_{test}$ **do**

15     Classification experts use $\delta_i^\star$ to first determine defect existence, then classify defect classes; detection experts use $\delta_j^\star$ to output label predictions;;

16     Ensemble predictions $\hat{y} = \sum_k w_k^\star \cdot \text{pred}_k$ and apply **decision rules** to obtain $\hat{y}_{\text{final}}$

17 **return** $\{\delta_k^\star\}, \{w_k^\star\}, \hat{y}_{\text{final}}$

---

Table 3: Distribution of samples across steel defect categories in the Severstal dataset, illustrating the pronounced long-tailed class imbalance in the training, validation, and test sets.

| | Numbers | Background | Defect-1 | Defect-2 | Defect-3 | Defect-4 |
|---|---|---|---|---|---|---|
| All Samples | 12568 | 5902(46.96%) | 897(7.14%) | 247(1.97%) | 5150(40.98%) | 801(6.37%) |
| Training Set | 10054 | 4719(46.94%) | 732(7.28%) | 202(2.01%) | 4117(40.95%) | 641(6.38%) |
| Validation Set | 1257 | 584(46.46%) | 86(6.84%) | 17(1.35%) | 531(42.24%) | 68(5.41%) |
| Test Set | 1257 | 599(47.65%) | 79(6.28%) | 28(2.22%) | 502(39.94%) | 92(7.32%) |

As illustrated in Fig. 7, the background images are not uniform gray surfaces but often contain pseudo-defects that resemble true defect patterns, thereby increasing the risk of false positives in classification. It is worth noting that the figure shows only a limited subset of background samples, whereas the full dataset exhibits even greater variability.

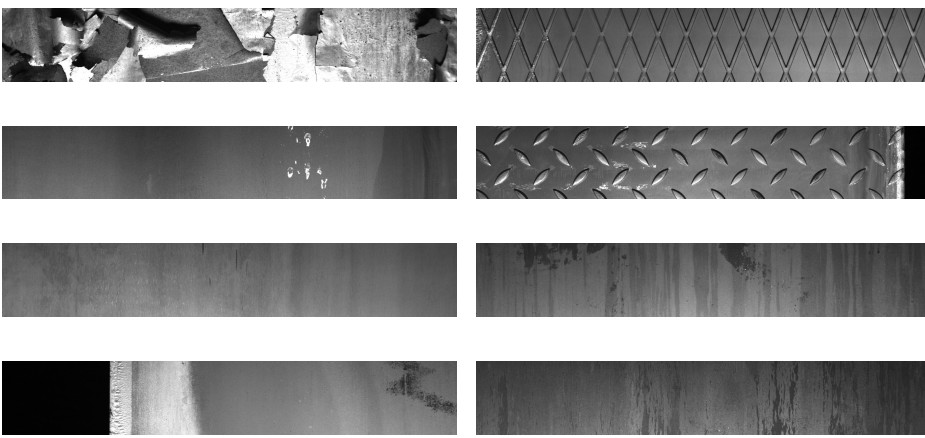

Figure 7: Representative background samples from the Severstal dataset, showing diverse surface textures and pseudo-defects that may lead to misclassification.

## C EXTENDED DISCUSSION ON TAIL-CLASS ENHANCEMENTS

For the classification experts, we present the detailed Accuracy results from Table 1, and as illustrated in Table 4, which allow us to observe the improvements for each defect class after applying different strategies to enhance the signal. Notably, we find that the tail-class 2 shows consistent improvement across all experts under the Two-Stage strategy, demonstrating that the Two-Stage design can effectively promote the recognition of tail(minority) class. In addition, the head-class 3 also exhibits substantial gains when using either the Two-Stage strategy alone or the Two-Stage strategy combined with augmentation.

Moreover, as shown in Fig. 8, we conducted a comparative analysis of Precision and Recall metrics to evaluate the impact of the Two-Stage strategy. The first five columns represent background images and defect classes 1–4, while the last column (Avg) indicates the average performance across the five categories. In the left panel, it can be observed that for most experts, the adoption of the Two-Stage strategy improves Precision across the majority of categories, with particularly notable gains for defect class 2. In contrast, the right panel demonstrates that the Two-Stage strategy offers negligible or no improvement in Recall, and in some cases even has a negative effect. This observation is consistent with the findings of Doyle et al. (2012), where hierarchical architectures were shown to propagate errors across stages. Although this strategy has some impact on Recall, the effect remains within a controllable range.

## D SUPPLEMENTARY ABLATION STUDY DETAILS

We conducted experiments with as many randomly selected diverse combinations as possible to ensure comprehensive evaluation. Table 5–Table 8 report the comparative results in terms of average Precision (Avg.$Pre$), average Recall (Avg.$Rec$), average F1-score (Avg.$F1$), and overall Accuracy (Over. $ACC$). Overall Accuracy is not the simple average of per-class accuracies, but rather the proportion of all correctly classified samples over the total number of samples. All classification models employed in this section were trained using the Two-Stage strategy. Within detection models, the DINO framework was omitted from ensemble candidates because of its substantial training cost, reflecting considerations of computational efficiency. All reported results were obtained under the setting where the weights of each expert were fixed to 1, while the confidence thresholds were optimized via grid search on the validation set.

Table 4: Per-class Accuracy ($ACC_i$) and average Accuracy ($ACC_{avg}$) results on the Severstal test set. $ACC_i$ ($i = 0$–4) denote the per-class Accuracy for background and four defect categories. Rows "texure", "contrastive", "augment", "2s", "2s_aug" show relative improvements (%) over the corresponding "1s".

| Expert | $ACC_0$ | $ACC_1$ | $ACC_2$ | $ACC_3$ | $ACC_4$ | $ACC_{avg}$ |
|---|---|---|---|---|---|---|
| Efficient_1s | 0.9412 | 0.9729 | 0.9857 | 0.9165 | 0.9880 | 0.9609 |
| Efficient_tex | -2.79% | +0.01% | **+0.41%** | -1.65% | -0.64% | -0.92% |
| Efficient_con | -1.44% | +0.25% | -0.08% | -0.17% | -0.31% | -0.35% |
| Efficient_aug | +0.41% | -0.89% | -0.32% | **+0.34%** | -0.15% | -0.14% |
| Efficient_2s | +0.07% | +0.41% | **+0.48%** | **+0.86%** | -0.64% | **+0.23%** |
| Efficient_2s_aug | +1.68% | -0.15% | +0.32% | **+2.17%** | +0.40% | **+0.85%** |
| ConvNeXt_1s | 0.9539 | 0.9857 | 0.9913 | 0.9220 | 0.9944 | 0.9695 |
| ConvNeXt_tex | -1.75% | -0.41% | **+0.23%** | +0.26% | -0.71% | -0.48% |
| ConvNeXt_con | -0.00% | -0.00% | -0.01% | -0.00% | +0.00% | -0.00% |
| ConvNeXt_aug | +0.83% | +0.00% | -0.08% | **+1.72%** | +0.08% | +0.48% |
| ConvNeXt_2s | -0.00% | -0.65% | **+0.23%** | **+2.07%** | -0.79% | **+0.14%** |
| ConvNeXt_2s_aug | +0.83% | +0.08% | -0.08% | **+3.54%** | -0.24% | **+0.78%** |
| ResNet101_1s | 0.9371 | 0.9737 | 0.9880 | 0.9133 | 0.9872 | 0.9599 |
| ResNet101_tex | +0.34% | +0.90% | **+0.32%** | +0.86% | +0.09% | +0.50% |
| ResNet101_con | +1.11% | +0.16% | +0.01% | +0.26% | -0.23% | +0.25% |
| ResNet101_aug | +0.17% | +0.82% | +0.17% | **+0.53%** | +0.41% | +0.42% |
| ResNet101_2s | +1.27% | +0.25% | **+0.16%** | **+2.01%** | +0.32% | **+0.77%** |
| ResNet101_2s_aug | +1.95% | +0.74% | +0.17% | **+3.13%** | +0.57% | **+1.28%** |
| VGG19_1s | 0.9618 | 0.9809 | 0.9905 | 0.9347 | 0.9936 | 0.9723 |
| VGG19_tex | -1.65% | +0.16% | **+0.07%** | -0.07% | -0.63% | -0.42% |
| VGG19_con | -0.42% | -0.16% | +0.31% | -0.76% | -0.31% | -0.26% |
| VGG19_aug | -0.17% | -0.00% | -0.16% | **+0.27%** | -0.16% | -0.05% |
| VGG19_2s | +0.08% | -0.33% | **+0.07%** | **+0.70%** | -0.40% | **+0.02%** |
| VGG19_2s_aug | -0.08% | -0.57% | +0.23% | **+1.37%** | -0.55% | **+0.07%** |

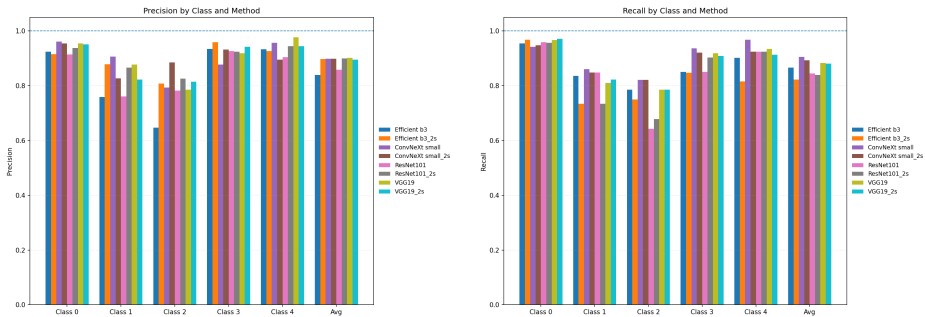

Figure 8: Performance comparison of different classification networks on the Severstal dataset under One-Stage and Two-Stage strategies: the left panel shows the average Precision across all categories, while the right panel presents the average Recall. Models without the "2s" suffix correspond to the One-Stage strategy, and class 0 denotes the background (no-defect) category.

In the ablation experiments conducted on the Severstal dataset, it was observed that the ensemble configuration comprising five experts (VGG + ResNet + ConvNeXt + YOLO11 + DetectoRS) achieved the best overall performance, attaining an average F1-score (*Avg. F*1) of 0.9489 and an overall Accuracy (*Over. ACC*) of 0.9610, which is significantly superior to other expert configurations.

Table 5: Performance comparison of two-expert ensemble models on the Severstal dataset.

| Model Combination | Avg.$Precision$ | Avg.$Recall$ | Avg.$F1-score$ | Overall.$Accuary$ |
|---|---|---|---|---|
| Efficient + ConvNeXt | 0.8968 | 0.9212 | 0.9086 | 0.9348 |
| VGG + ConvNeXt | 0.9069 | 0.9309 | 0.9185 | 0.9395 |
| VGG + Efficient | 0.9112 | 0.9242 | 0.9176 | 0.9403 |
| ResNet + ConvNeXt | 0.8947 | 0.9342 | 0.9139 | 0.9292 |
| DetectoRS + YOLO11 | 0.7899 | 0.8704 | 0.8062 | 0.7963 |
| DetectoRS + YOLO8 | 0.7974 | 0.8569 | 0.8046 | 0.7995 |
| ConvNeXt + YOLO11 | 0.9105 | 0.9366 | 0.9233 | 0.9340 |
| Efficient + YOLO11 | 0.9184 | 0.9449 | **0.9308** | 0.9395 |
| VGG + YOLO11 | 0.9082 | 0.9538 | 0.9299 | 0.9340 |

Table 6: Performance comparison of three-expert ensemble models on the Severstal dataset.

| Model Combination | Avg.$Pre$ | Avg.$Rec$ | Avg.$F1$ | Over.$ACC$ |
|---|---|---|---|---|
| ResNet + Efficient + ConvNeXt | 0.9327 | 0.9024 | 0.9169 | 0.9435 |
| ResNet + Efficient + VGG | 0.9329 | 0.9096 | 0.9208 | 0.9411 |
| ConvNeXt + Efficient + VGG | 0.9232 | 0.9137 | 0.9183 | 0.9403 |
| YOLO11 + Efficient + VGG | 0.9449 | 0.9256 | 0.9349 | 0.9443 |
| YOLO11 + Efficient + ConvNeXt | 0.9394 | 0.9166 | 0.9275 | 0.9451 |
| YOLO11 + VGG + ConvNeXt | 0.9397 | 0.9359 | 0.9377 | 0.9507 |
| YOLO11 + VGG + DetectoRS | 0.9448 | 0.9212 | 0.9326 | 0.9507 |
| YOLO11 + VGG + ResNet | 0.9448 | 0.9300 | 0.9372 | 0.9467 |
| YOLO11 + ConvNeXt + ResNet | 0.9574 | 0.9212 | **0.9387** | 0.9554 |

Table 7: Performance comparison of four-expert ensemble models on the Severstal dataset.

| Model Combination | Avg.$Prec$ | Avg.$Rec$ | Avg.$F1$ | Over.$ACC$ |
|---|---|---|---|---|
| VGG + ResNet + ConvNeXt + Efficient | 0.9256 | 0.9224 | 0.9240 | 0.9451 |
| VGG + ResNet + YOLO11 + DetectoRS | 0.9484 | 0.9481 | **0.9480** | 0.9531 |
| ConvNeXt + Efficient + YOLO11 + DetectoRS | 0.9343 | 0.9322 | 0.9332 | 0.9459 |
| VGG + ResNet + ConvNeXt + YOLO11 | 0.9440 | 0.9392 | 0.9415 | 0.9547 |
| VGG + ResNet + Efficient + YOLO11 | 0.9495 | 0.9447 | 0.9469 | 0.9547 |
| VGG + Efficient + ConvNeXt + YOLO11 | 0.9329 | 0.9354 | 0.9341 | 0.9459 |

# E    PRECISION–RECALL PERFORMANCE AND STABILITY ANALYSIS ON THE SEVERSTAL TEST SET

In this section, we also report the Precision and Recall on the Severstal dataset. As shown in Table 9–Table 10, it is evident that our method (*Ours_2s*) consistently outperforms other approaches across individual defect classes. Our method achieves the highest performance ($P_{avg}$=0.9547, $R_{avg}$=0.9455), surpassing the cutting-edge YOLO11 model by approximately 2%. These findings highlight the superior accuracy of our method.

Table 8: Performance comparison of five-expert ensemble models on the Severstal dataset.

| Model Combination | Avg.$Pre$ | Avg.$Rec$ | Avg.$F1$ | Over.$ACC$ |
|---|---|---|---|---|
| VGG + ResNet + ConvNeXt + Efficient + YOLO11 | 0.9496 | 0.9283 | 0.9386 | 0.9539 |
| VGG + ResNet + ConvNeXt + Efficient + DetectoRS | 0.9457 | 0.9380 | 0.9417 | 0.9523 |
| VGG + ResNet + Cascade + YOLO11 + DetectoRS | 0.9574 | 0.9322 | 0.9444 | 0.9570 |
| VGG + ResNet + ConvNeXt + YOLO11 + DetectoRS | 0.9599 | 0.9384 | **0.9489** | 0.9610 |
| VGG + ConvNeXt + Efficient + YOLO11 + DetectoRS | 0.9450 | 0.9275 | 0.9361 | 0.9515 |
| ResNet + ConvNeXt + Efficient + YOLO11 + DetectoRS | 0.9354 | 0.9312 | 0.9332 | 0.9443 |

Table 9: Per-class Precision ($P_i$) and average Precision ($P_{avg}$) results on the Severstal test set. Here, $P_i$ (i=0–4) denote the per-class Precision for background and four defect categories. Methodological variants:(1s)One-Stage strategy in the classification branch; (2s)Two-Stage strategy;(2s_aug)Two-Stage strategy with data augmentation techniques.

| Expert | $P_0$ | $P_1$ | $P_2$ | $P_3$ | $P_4$ | $P_{\mathrm{avg}}$ |
|---|---|---|---|---|---|---|
| Efficient_1s | 0.9241 | 0.7586 | 0.6471 | 0.9344 | 0.9326 | 0.8394 |
| ConvNeXt_1s | 0.9608 | 0.9067 | 0.7931 | 0.8769 | 0.9570 | 0.8989 |
| ResNet101_1s | 0.9140 | 0.7614 | 0.7826 | 0.9262 | 0.9043 | 0.8577 |
| VGG19_1s | 0.9539 | 0.8767 | 0.7857 | 0.9183 | 0.9773 | 0.9024 |
| Efficient_2s | 0.9148 | 0.8788 | 0.8077 | 0.9594 | 0.9259 | 0.8973 |
| ConvNeXt_2s | 0.9546 | 0.8272 | 0.8846 | 0.9315 | 0.8947 | 0.8985 |
| ResNet101_2s | 0.9378 | 0.8657 | 0.8261 | 0.9245 | 0.9444 | 0.8997 |
| VGG19_2s | 0.9510 | 0.8228 | 0.8148 | 0.9421 | 0.9438 | 0.8949 |
| Efficient_2s_aug | 0.9690 | 0.7792 | 0.6750 | 0.9271 | 0.9457 | 0.8592 |
| ConvNeXt_2s_aug | 0.9742 | 0.8875 | 0.8077 | 0.9389 | 0.9457 | 0.9108 |
| ResNet101_2s_aug | 0.9458 | 0.8767 | **0.9412** | 0.9633 | 0.9770 | 0.9408 |
| VGG19_2s_aug | 0.9599 | 0.7791 | 0.8519 | 0.9291 | 0.9529 | 0.8946 |
| Faster R-CNN | 0.9561 | 0.6765 | 0.9200 | 0.6477 | 0.4314 | 0.7263 |
| Cascade R-CNN | 0.9420 | 0.6389 | 0.6875 | 0.5751 | 0.4147 | 0.6516 |
| DetectoRS | 0.9397 | 0.8608 | 0.8846 | 0.8125 | 0.4759 | 0.7947 |
| DINO | 0.9543 | 0.9114 | 0.9167 | 0.9504 | 0.9583 | 0.9382 |
| YOLO8 | 0.9692 | 0.8846 | 0.9167 | 0.9110 | 0.9468 | 0.9257 |
| YOLO11 | 0.9560 | 0.9189 | 0.9231 | 0.9378 | 0.9884 | 0.9448 |
| Voting_7 | 0.9605 | 0.9324 | 0.9130 | 0.9386 | 0.9888 | 0.9467 |
| **Ours_1s** | 0.9636 | 0.9200 | 0.9130 | 0.9371 | 0.9888 | 0.9445 |
| **Ours_2s** | **0.9917** | **0.9324** | 0.8966 | **0.9639** | 0.9889 | **0.9547** |
| **Ours_2s_aug** | 0.9819 | 0.9200 | 0.8800 | 0.9546 | **0.9890** | 0.9451 |

Table 10: Per-class Recall ($R_i$) and average Recall ($R_{avg}$) results on the Severstal test set.

| Expert | $R_0$ | $R_1$ | $R_2$ | $R_3$ | $R_4$ | $R_{\text{avg}}$ |
|---|---|---|---|---|---|---|
| Efficient_1s | 0.9549 | 0.8354 | 0.7857 | 0.8506 | 0.9022 | 0.8658 |
| ConvNeXt_1s | 0.9416 | 0.8608 | 0.8214 | 0.9363 | 0.9674 | 0.9055 |
| ResNet101_1s | 0.9583 | 0.8481 | 0.6429 | 0.8506 | 0.9239 | 0.8448 |
| VGG19_1s | 0.9666 | 0.8101 | 0.7857 | 0.9183 | 0.9348 | 0.8831 |
| Efficient_2s | 0.9683 | 0.7342 | 0.7500 | 0.8466 | 0.8152 | 0.8229 |
| ConvNeXt_2s | 0.9482 | 0.8481 | 0.8214 | 0.9203 | 0.9239 | 0.8924 |
| ResNet101_2s | 0.9566 | 0.7342 | 0.6786 | 0.9024 | 0.9239 | 0.8391 |
| VGG19_2s | 0.9716 | 0.8228 | 0.7857 | 0.9084 | 0.9130 | 0.8803 |
| Efficient_2s_aug | 0.9399 | 0.7595 | **0.9643** | 0.9124 | 0.9457 | 0.9043 |
| ConvNeXt_2s_aug | 0.9449 | 0.8987 | 0.7500 | 0.9482 | 0.9457 | 0.8975 |
| ResNet101_2s_aug | 0.9616 | 0.8101 | 0.5714 | 0.8884 | 0.9239 | 0.8311 |
| VGG19_2s_aug | 0.9583 | 0.8481 | 0.8214 | 0.9402 | 0.8804 | 0.8897 |
| Faster R-CNN | 0.4725 | 0.8734 | 0.8214 | **0.9741** | 0.9565 | 0.8196 |
| Cascade R-CNN | 0.3255 | 0.8734 | 0.7857 | 0.9681 | 0.9783 | 0.7862 |
| DetectoRS | 0.6761 | 0.8608 | 0.8214 | 0.9323 | 0.9674 | 0.8516 |
| DINO | 0.9766 | **0.9114** | 0.7857 | 0.9163 | **1.0000** | 0.9180 |
| YOLO8 | 0.9466 | 0.8734 | 0.7857 | 0.9582 | 0.9674 | 0.9063 |
| YOLO11 | 0.9800 | 0.8608 | 0.8571 | 0.9303 | 0.9239 | 0.9104 |
| Voting_7 | 0.9750 | 0.8734 | 0.7500 | 0.9442 | 0.9565 | 0.8998 |
| **Ours_1s** | 0.9733 | 0.8734 | 0.7500 | 0.9502 | 0.9565 | 0.9007 |
| **Ours_2s** | **1.0000** | 0.8734 | 0.9286 | 0.9582 | 0.9674 | **0.9455** |
| **Ours_2s_aug** | 0.9983 | 0.8734 | 0.7857 | 0.9641 | 0.9783 | 0.9200 |

To assess the robustness and stability of the proposed method, we repeat experiments on the Severstal dataset using five different random seeds. For each seed, we compute per-class and averaged metrics for Precision, Recall, F1-score, and Accuracy. We then aggregate the five runs and report the results as mean (sd), where "sd" denotes the standard deviation across seeds. The detailed outcomes are summarized in Table 11–Table 14, corresponding to Precision, Recall, F1-score, and Accuracy, respectively.

From these numerical results, our method (Ours_2s) achieves the best averaged performance across Precision, Recall, F1-score, and Accuracy, while maintaining notably low standard deviations (around 1%). This demonstrates that the proposed multi-expert ensemble is both effective and robust under different random initializations, consistently delivering strong results on the Severstal test set. Importantly, these results highlight that the performance improvements arise from our novel framework design and adaptive ensemble mechanism, which jointly ensure both stability and reliability rather than relying on randomness.

Table 11: Per-class and averaged Precision on the Severstal dataset over five random seeds, reported as mean (sd).

| Expert | $P_0$ | $P_1$ | $P_2$ | $P_3$ | $P_4$ | $P_{\text{avg}}$ |
|---|---|---|---|---|---|---|
| Efficient_2s | 0.935 (0.014) | 0.867 (0.034) | 0.842 (0.031) | 0.959 (0.010) | 0.908 (0.038) | 0.902 (0.017) |
| ConvNeXt_2s | 0.953 (0.008) | 0.867 (0.033) | 0.901 (0.018) | 0.943 (0.015) | 0.947 (0.033) | 0.922 (0.014) |
| ResNet101_2s | 0.948 (0.013) | 0.796 (0.042) | 0.835 (0.046) | 0.932 (0.017) | 0.947 (0.016) | 0.892 (0.009) |
| VGG19_2s | 0.956 (0.003) | 0.832 (0.032) | 0.857 (0.051) | 0.945 (0.007) | 0.968 (0.015) | 0.912 (0.010) |
| Faster R-CNN | 0.943 (0.025) | 0.675 (0.091) | 0.830 (0.038) | 0.648 (0.028) | 0.455 (0.020) | 0.710 (0.023) |
| YOLO8 | 0.960 (0.017) | 0.857 (0.025) | 0.875 (0.047) | 0.903 (0.017) | 0.963 (0.015) | 0.912 (0.015) |
| YOLO11 | 0.962 (0.014) | 0.894 (0.016) | 0.897 (0.041) | 0.913 (0.019) | 0.969 (0.018) | 0.927 (0.014) |
| Ours_2s | 0.988 (0.006) | 0.918 (0.015) | 0.932 (0.031) | 0.955 (0.008) | 0.980 (0.014) | 0.955 (0.011) |

Table 12: Per-class and averaged Recall on the Severstal dataset over five random seeds, reported as mean (sd).

| Expert | $R_0$ | $R_1$ | $R_2$ | $R_3$ | $R_4$ | $R_{\text{avg}}$ |
|---|---|---|---|---|---|---|
| Efficient_2s | 0.964 (0.007) | 0.775 (0.052) | 0.700 (0.109) | 0.866 (0.022) | 0.837 (0.041) | 0.828 (0.029) |
| ConvNeXt_2s | 0.937 (0.015) | 0.846 (0.011) | 0.779 (0.030) | 0.914 (0.016) | 0.893 (0.039) | 0.874 (0.012) |
| ResNet101_2s | 0.949 (0.012) | 0.810 (0.043) | 0.736 (0.060) | 0.900 (0.025) | 0.878 (0.049) | 0.855 (0.029) |
| VGG19_2s | 0.960 (0.008) | 0.820 (0.005) | 0.736 (0.048) | 0.912 (0.004) | 0.900 (0.013) | 0.866 (0.010) |
| Faster R-CNN | 0.497 (0.048) | 0.856 (0.032) | 0.700 (0.048) | 0.958 (0.016) | 0.957 (0.018) | 0.793 (0.014) |
| YOLO8 | 0.948 (0.013) | 0.858 (0.020) | 0.864 (0.089) | 0.947 (0.022) | 0.952 (0.011) | 0.914 (0.024) |
| YOLO11 | 0.963 (0.006) | 0.866 (0.035) | 0.843 (0.043) | 0.943 (0.018) | 0.941 (0.009) | 0.911 (0.019) |
| Ours_2s | 0.999 (0.002) | 0.873 (0.009) | 0.879 (0.054) | 0.959 (0.009) | 0.963 (0.010) | 0.935 (0.014) |

Table 13: Per-class and averaged F1-score on the Severstal dataset over five random seeds, reported as mean (sd).

| Expert | $F1_0$ | $F1_1$ | $F1_2$ | $F1_3$ | $F1_4$ | $F1_{\text{avg}}$ |
|---|---|---|---|---|---|---|
| Efficient_2s | 0.949 (0.006) | 0.817 (0.024) | 0.759 (0.060) | 0.910 (0.008) | 0.870 (0.016) | 0.861 (0.011) |
| ConvNeXt_2s | 0.945 (0.005) | 0.856 (0.012) | 0.835 (0.013) | 0.928 (0.002) | 0.918 (0.014) | 0.896 (0.002) |
| ResNet101_2s | 0.949 (0.006) | 0.802 (0.029) | 0.780 (0.021) | 0.915 (0.008) | 0.910 (0.024) | 0.871 (0.012 |
| VGG19_2s | 0.958 (0.004) | 0.826 (0.015) | 0.789 (0.015) | 0.928 (0.002) | 0.932 (0.008) | 0.887 (0.006) |
| Faster R-CNN | 0.649 (0.037) | 0.749 (0.047) | 0.759 (0.042) | 0.773 (0.016) | 0.616 (0.018) | 0.709 (0.020) |
| YOLO8 | 0.954 (0.004) | 0.857 (0.012) | 0.865 (0.043) | 0.924 (0.004) | 0.957 (0.012) | 0.912 (0.011) |
| YOLO11 | 0.962 (0.005) | 0.879 (0.011) | 0.868 (0.014) | 0.928 (0.005) | 0.955 (0.008) | 0.918 (0.004) |
| Ours_2s | 0.994 (0.004) | 0.895 (0.005) | 0.904 (0.037) | 0.957 (0.004) | 0.972 (0.002) | 0.944 (0.009) |

Table 14: Per-class and averaged Accuracy on the Severstal dataset over five random seeds, reported as mean (sd).

| Expert | $ACC_0$ | $ACC_1$ | $ACC_2$ | $ACC_3$ | $ACC_4$ | $ACC_{\text{avg}}$ |
|---|---|---|---|---|---|---|
| Efficient_2s | 0.951 (0.006) | 0.978 (0.002) | 0.990 (0.001) | 0.932 (0.005) | 0.982 (0.002) | 0.967 (0.002) |
| ConvNeXt_2s | 0.948 (0.005) | 0.982 (0.002) | 0.993 (0.000) | 0.943 (0.002) | 0.988 (0.002) | 0.971 (0.001) |
| ResNet101_2s | 0.951 (0.005) | 0.975 (0.004) | 0.991 (0.001) | 0.934 (0.006) | 0.987 (0.003) | 0.968 (0.002) |
| VGG19_2s | 0.960 (0.003) | 0.978 (0.002) | 0.991 (0.000) | 0.944 (0.002) | 0.990 (0.001) | 0.973 (0.001) |
| Faster R-CNN | 0.432 (0.053) | 0.856 (0.032) | 0.700 (0.048) | 0.958 (0.016) | 0.957 (0.018) | 0.780 (0.013) |
| YOLO8 | 0.956 (0.004) | 0.982 (0.002) | 0.994 (0.002) | 0.938 (0.003) | 0.994 (0.002) | 0.973 (0.002) |
| YOLO11 | 0.964 (0.005) | 0.985 (0.001) | 0.994 (0.001) | 0.941 (0.004) | 0.993 (0.001) | 0.976 (0.001) |
| Ours_2s | 0.994 (0.004) | 0.987 (0.001) | 0.996 (0.002) | 0.966 (0.003) | 0.996 (0.000) | 0.988 (0.001) |

Besides, we recorded the accuracies of both stages in the two-phase classification model in Table 15. The results indicate that, for all models, the first stage consistently achieves higher accuracy than the second stage, as it involves a simpler binary classification task, whereas the second stage requires distinguishing among multiple classes. This demonstrates that first identifying the presence of a defect and subsequently classifying its specific type is more effective than performing direct multi-class classification. Simplifying a complex task into sequential sub-tasks can therefore lead to improved overall accuracy.

## F    SUPPLEMENTARY INFERENCE EFFICIENCY ON THE SEVERTAL DATA

As shown in Table 16, the Cascade, DetectoRS and DINO exhibit the lowest inference efficiency among all evaluated models. In comparison, the efficiency of our proposed method is primarily constrained by the slowest expert within the ensemble, namely the Faster R-CNN model.

Table 15: Performance of Two-Stage Classification on the Severstal dataset

| Method | First-stage classifier | | | | Second-stage classifier | | | |
|---|---|---|---|---|---|---|---|---|
| | P | R | F1 | Acc | P | R | F1 | Acc |
| Efficient | 0.9695 | 0.9179 | 0.9430 | 0.9419 | 0.8973 | 0.8229 | 0.8570 | 0.9631 |
| ConvNeXt | 0.9532 | 0.9590 | 0.9561 | 0.9539 | 0.8985 | 0.8924 | 0.8952 | 0.9709 |
| ResNet101 | 0.9444 | 0.9559 | 0.9502 | 0.9475 | 0.8997 | 0.8391 | 0.8668 | 0.9673 |
| VGG19 | 0.9736 | 0.9544 | 0.9639 | 0.9626 | 0.8949 | 0.8803 | 0.8874 | 0.9725 |

Table 16: Inference Efficiency and GPU Memory Usage on the Severtal data

| Method | Avg. Time (s/img) | Throughput (img/s) | GPU Mem (MB/img) |
|---|---|---|---|
| Efficient | 0.0008 | 1292.22 | 18.59 |
| ConvNeXt | 0.0011 | 885.65 | 24.48 |
| ResNet101 | 0.0005 | 1689.23 | 21.40 |
| VGG19 | 0.001 | 1032.60 | 71.12 |
| Faster R-CNN | 0.028 | 35.26 | 282.47 |
| Cascade R-CNN | 0.097 | 10.36 | 446.02 |
| DetectoRS | 0.184 | 5.42 | 554.45 |
| YOLO8 | 0.014 | 72.81 | 104.80 |
| YOLO11 | 0.025 | 39.73 | 151.83 |
| DINO | 0.072 | 13.88 | 284.72 |

## G  EXPERIMENTAL RESULTS ON THE NEU DATASET

For the NEU dataset, DetectoRS, DINO, and Cascade R-CNN were excluded due to excessive inference latency, and the remaining experts were integrated. As reported in Table 17, our proposed ensemble consistently attains superior results across key metrics, yielding $F1_{\text{avg}} = 0.9827$, and $ACC_{\text{avg}} = 0.9926$. These results demonstrate that the proposed framework achieves a more favorable balance across multiple evaluation metrics compared with existing baselines. Notably, while DetectoRS attains competitive accuracy, our approach offers substantially improved efficiency, thereby enhancing practical applicability.

Table 17: Average Precision, Recall, F1-score, and ACC of different methods on the NEU test set.

| Expert | $P_{\text{avg}}$ | $R_{\text{avg}}$ | $F1_{\text{avg}}$ | $ACC_{\text{avg}}$ |
|---|---|---|---|---|
| Efficient b3 | 0.9899 | 0.9491 | 0.9677 | 0.9870 |
| ConvNeXt small | 0.9783 | 0.9783 | 0.9783 | 0.9907 |
| ResNet101 | 0.9710 | 0.9518 | 0.9605 | 0.9843 |
| VGG19 | **0.9915** | 0.9591 | 0.9746 | 0.9898 |
| Faster R-CNN | 0.9555 | **0.9912** | 0.9724 | 0.9879 |
| Cascade R-CNN | 0.9745 | 0.9734 | 0.9739 | 0.9889 |
| DetectoRS | 0.9832 | 0.9823 | **0.9827** | **0.9926** |
| DINO | 0.9862 | 0.9749 | 0.9801 | 0.9917 |
| **Ours** | 0.9832 | 0.9823 | **0.9827** | **0.9926** |

Table 18–Table 21 present the detailed experimental results on the NEU dataset, covering all six classes as well as their averaged performance, and reporting four evaluation metrics: Precision, Recall, F1-score, and Accuracy.

## H  FUTURE WORK

Several promising research directions emerge from this work. First, establishing theoretical foundations that explain the conditions under which our multi-expert ensemble achieves superior performance would provide deeper insights and guide systematic expert selection strategies. Second, extending the framework to support incremental learning with continuous data streams represents a crucial step toward adaptive systems that can incorporate new defect patterns without requiring

Table 18: Per-class Precision ($P_i$) and average Precision ($P_{avg}$) results on the NEU test set.

| Expert | $P_1$ | $P_2$ | $P_3$ | $P_4$ | $P_5$ | $P_6$ | $P_{\text{avg}}$ |
|---|---|---|---|---|---|---|---|
| Efficient b3 | 1.0000 | 0.9697 | 0.9697 | 1.0000 | 1.0000 | 1.0000 | 0.9899 |
| ConvNeXt small | 1.0000 | 0.9268 | 0.9429 | 1.0000 | 1.0000 | 1.0000 | 0.9783 |
| ResNet101 | 0.9375 | 0.9211 | 0.9677 | 1.0000 | 1.0000 | 1.0000 | 0.9710 |
| VGG19 | 1.0000 | 0.9487 | 1.0000 | 1.0000 | 1.0000 | 1.0000 | **0.9915** |
| Faster R-CNN | 1.0000 | 0.8511 | 0.9444 | 0.9375 | 1.0000 | 1.0000 | 0.9555 |
| Cascade R-CNN | 1.0000 | 0.9024 | 0.9444 | 1.0000 | 1.0000 | 1.0000 | 0.9745 |
| DetectoRS | 1.0000 | 0.9286 | 0.9706 | 1.0000 | 1.0000 | 1.0000 | 0.9832 |
| DINO | 1.0000 | 0.9730 | 0.9444 | 1.0000 | 1.0000 | 1.0000 | 0.9862 |
| **Ours** | 1.0000 | 0.9286 | 0.9706 | 1.0000 | 1.0000 | 1.0000 | 0.9832 |

Table 19: Per-class Recall ($R_i$) and average Recall ($R_{avg}$) results on the NEU test set.

| Expert | $R_1$ | $R_2$ | $R_3$ | $R_4$ | $R_5$ | $R_6$ | $R_{\text{avg}}$ |
|---|---|---|---|---|---|---|---|
| Efficient b3 | 1.0000 | 0.7805 | 0.9143 | 1.0000 | 1.0000 | 1.0000 | 0.9491 |
| ConvNeXt small | 1.0000 | 0.9268 | 0.9429 | 1.0000 | 1.0000 | 1.0000 | 0.9783 |
| ResNet101 | 1.0000 | 0.8537 | 0.8571 | 1.0000 | 1.0000 | 1.0000 | 0.9518 |
| VGG19 | 1.0000 | 0.9024 | 0.8857 | 1.0000 | 1.0000 | 0.9667 | 0.9591 |
| Faster R-CNN | 1.0000 | 0.9756 | 0.9714 | 1.0000 | 1.0000 | 1.0000 | **0.9912** |
| Cascade R-CNN | 1.0000 | 0.9024 | 0.9714 | 1.0000 | 0.9667 | 1.0000 | 0.9734 |
| DetectoRS | 1.0000 | 0.9512 | 0.9429 | 1.0000 | 1.0000 | 1.0000 | 0.9823 |
| DINO | 1.0000 | 0.8780 | 0.9714 | 1.0000 | 1.0000 | 1.0000 | 0.9749 |
| **Ours** | 1.0000 | 0.9512 | 0.9429 | 1.0000 | 1.0000 | 1.0000 | 0.9823 |

Table 20: Per-class F1-score ($F1_i$) and average F1-score ($F1_{avg}$) results on the NEU test set.

| Expert | $F1_1$ | $F1_2$ | $F1_3$ | $F1_4$ | $F1_5$ | $F1_6$ | $F1_{\text{avg}}$ |
|---|---|---|---|---|---|---|---|
| Efficient b3 | 1.0000 | 0.8649 | 0.9412 | 1.0000 | 1.0000 | 1.0000 | 0.9677 |
| ConvNeXt small | 1.0000 | 0.9268 | 0.9429 | 1.0000 | 1.0000 | 1.0000 | 0.9783 |
| ResNet101 | 0.9677 | 0.8861 | 0.9091 | 1.0000 | 1.0000 | 1.0000 | 0.9605 |
| VGG19 | 1.0000 | 0.9250 | 0.9394 | 1.0000 | 1.0000 | 0.9831 | 0.9746 |
| Faster R-CNN | 1.0000 | 0.9091 | 0.9577 | 0.9677 | 1.0000 | 1.0000 | 0.9724 |
| Cascade R-CNN | 1.0000 | 0.9024 | 0.9577 | 1.0000 | 0.9831 | 1.0000 | 0.9739 |
| DetectoRS | 1.0000 | 0.9398 | 0.9565 | 1.0000 | 1.0000 | 1.0000 | **0.9827** |
| DINO | 1.0000 | 0.9231 | 0.9577 | 1.0000 | 1.0000 | 1.0000 | 0.9801 |
| **Ours** | 1.0000 | 0.9398 | 0.9565 | 1.0000 | 1.0000 | 1.0000 | **0.9827** |

Table 21: Per-class Accuracy ($ACC_i$) and average Accuracy ($ACC_{avg}$) results on the NEU test set.

| Expert | $ACC_1$ | $ACC_2$ | $ACC_3$ | $ACC_4$ | $ACC_5$ | $ACC_6$ | $ACC_{\text{avg}}$ |
|---|---|---|---|---|---|---|---|
| Efficient b3 | 1.0000 | 0.9445 | 0.9778 | 1.0000 | 1.0000 | 1.0000 | 0.9870 |
| ConvNeXt small | 1.0000 | 0.9667 | 0.9777 | 1.0000 | 1.0000 | 1.0000 | 0.9907 |
| ResNet101 | 0.9889 | 0.9500 | 0.9667 | 1.0000 | 1.0000 | 1.0000 | 0.9843 |
| VGG19 | 1.0000 | 0.9667 | 0.9778 | 1.0000 | 1.0000 | 0.9944 | 0.9898 |
| Faster R-CNN | 1.0000 | 0.9555 | 0.9833 | 0.9889 | 1.0000 | 1.0000 | 0.9879 |
| Cascade R-CNN | 1.0000 | 0.9556 | 0.9833 | 1.0000 | 0.9944 | 1.0000 | 0.9889 |
| DetectoRS | 1.0000 | 0.9723 | 0.9833 | 1.0000 | 1.0000 | 1.0000 | **0.9926** |
| DINO | 1.0000 | 0.9667 | 0.9833 | 1.0000 | 1.0000 | 1.0000 | 0.9917 |
| **Ours** | 1.0000 | 0.9722 | 0.9833 | 1.0000 | 1.0000 | 1.0000 | **0.9926** |

complete model retraining. Additionally, integrating knowledge from large foundation models, including large language models, presents an exciting opportunity to further enhance defect detection accuracy. These models could potentially contribute semantic understanding, contextual reasoning, or transfer learning capabilities that complement our ensemble's visual pattern recognition strengths.

# I   USE OF LARGE LANGUAGE MODELS

In accordance with the ICLR 2026 policy on the disclosure of large language model (LLM) usage, we provide the following details regarding the role of LLMs in the preparation of this paper.

Large language models were employed in Writing Polish: An LLM was used to aid in improving the clarity, grammar, and readability of the manuscript text. All technical ideas, analysis, and conclusions remain the authors' original work.

The use of LLMs was restricted to the above purpose. No part of the conceptualization, experimental design, interpretation of results was produced by an LLM.

