# OpenReview forum: "A Multi-Expert Ensemble Model for Long-Tailed Steel Surface Defect Detection"
_ICLR.cc/2026/Conference — ICLR 2026 Conference Withdrawn Submission_

### Official Review · Reviewer_hTzw · 2025-10-26

**Soundness:** 3
**Presentation:** 2
**Contribution:** 2
**Rating:** 2
**Confidence:** 4

**Summary:**

The paper proposes a recipe for how to train a MoE-style ensemble of methods. The ensemble consists of classification networks such as VGG and ResNet and object detection networks such as YOLO and Faster R-CNN. The method is evaluated on two datasets: Severstal and NEU. The method improves upon previous results.

**Strengths:**

- Ensembling methods in a MoE-style network makes sense.
- Additionally, combining both classification networks and object detection networks makes sense.

**Weaknesses:**

- The paper is hard to follow at times and could use improved presentation.
- No “newer” methods, such as RF-DETR [1] and RADIO [2], were evaluated putting into question whether such ensembling is even required
- No qualitative examples are presented in the paper
- It is unclear which models are used in the final ensemble. All of the proposed (4 classification and 6 detection)?
- The ablation study does not validate that this is indeed better than a straight-up combination of networks

[1]  Robinson, I., Robicheaux, P., Popov, M., Ramanan, D., & Peri, N.. (2025). RF-DETR.

[2] Heinrich, G., Ranzinger, M., Yin, H., Lu, Y., Kautz, J., Tao, A., ... & Molchanov, P. (2025). Radiov2. 5: Improved baselines for agglomerative vision foundation models. In Proceedings of the Computer Vision and Pattern Recognition Conference (pp. 22487-22497).

**Questions:**

I have several questions. I have sorted them from most problematic to least problematic.

1. How does the model perform when equal weights are assigned to each model?
2. What is the performance of newer methods such as RF-DETR and RADIO?
3. Which models are used in the final ensemble?

---

> ### Author Response · Authors · 2025-11-21
> **Response to Reviewer hTzw**
>
> Thank you for your insightful feedback
>
> W1: It is unclear which models are included in the final ensemble.
>
> answer: In Section 5.4, Experimental Results on the Severstal Dataset, we clarify in the opening sentence that our ensemble model excludes experts with high inference costs—namely DetectoRS, DINO, and Cascade R-CNN. The remaining seven experts constitute the set used in our ensemble framework described in the paper.
>
> W2+Q1: How does the model perform when equal weights are assigned to each model? The ablation study does not demonstrate that the ensemble is superior to simple model combination.
>
> answer: We have added the voting results to the main text. Our findings indicate that simple model voting does not effectively improve the F1-score for the second tail class. In contrast, the proposed ensemble method achieves a clear and meaningful improvement in the F1-score for this class.
>
> Q2: Why were newer methods such as RF-DETR and RADIO not included in the comparisons?
>
> answer: As our literature coverage is still limited, we had not yet examined the two methods you mentioned. We appreciate your recommendation and will study and learn from them. In addition, RF-DETR is an extension of the DETR family, and DINO—one of the baselines we compare against—is also a DETR-based method. We believe that the models already included in our comparison are sufficiently representative. Even if RF-DETR or RADIO achieve strong performance, incorporating them into our comparison—and potentially into our expert ensemble—would only further improve the overall accuracy of our method. We will continue to monitor newly proposed approaches and expand the expert pool of our framework accordingly.
>
> Q3: Which specific models are included in the final ensemble?
>
> answer: In Section 5.4, Experimental Results on the Severstal Dataset, we state in the opening sentence that our ensemble model excludes models with high inference costs—specifically DetectoRS, DINO, and Cascade R-CNN. The remaining seven experts constitute the ensemble used in our proposed framework.

---

> > ### Comment · Reviewer_hTzw · 2025-11-24
> > **Official Comment by Reviewer hTzw**
> >
> > I thank the authors for their responses.
> >
> > Some of my concerns were alleviated (specifically, which models are included in the final ensemble and the addition of the ablation with equal weights).
> >
> > However, my concern about the performance of newer methods is not sufficiently answered. There is still a possibility that the problems they are trying to solve are non-existent with newer and stronger baselines. For instance some SSL-based methods, such as DINOv2[1] and RADIO[2] can be easily trained as a classifier, and newer and stronger object detection methods, such as GroundingDINO[3], can be included. Without these results, it is questionable whether this tail-class underperformance even exists with stronger baselines and whether the proposed ensembling method improves their performance.
> >
> > Until this concern is sufficiently answered, I cannot increase my original rating. Due to that, I am still inclined to recommend rejection.
> >
> > Additionally, the presentation remains subpar and requires improvement. Just to list a few: The text in Figures 5 and 6 is too small, and Table 2 goes beyond the margins; the description of how the ensemble is calculated is way too brief. There are many more, but I will not elaborate further.
> >
> > [1] Oquab, M., Darcet, T., Moutakanni, T., Vo, H., Szafraniec, M., Khalidov, V., ... & Bojanowski, P. (2023). Dinov2: Learning robust visual features without supervision. arXiv preprint arXiv:2304.07193.
> >
> > [2] Heinrich, G., Ranzinger, M., Yin, H., Lu, Y., Kautz, J., Tao, A., ... & Molchanov, P. (2025). Radiov2. 5: Improved baselines for agglomerative vision foundation models. In Proceedings of the Computer Vision and Pattern Recognition Conference (pp. 22487-22497).
> >
> > [3] Liu, S., Zeng, Z., Ren, T., Li, F., Zhang, H., Yang, J., ... & Zhang, L. (2024, September). Grounding dino: Marrying dino with grounded pre-training for open-set object detection. In European conference on computer vision (pp. 38-55). Cham: Springer Nature Switzerland.

---

### Official Review · Reviewer_pepC · 2025-10-27

**Soundness:** 2
**Presentation:** 1
**Contribution:** 1
**Rating:** 2
**Confidence:** 5

**Summary:**

This paper proposes a multi-expert ensemble framework combining classification and detection models to address the challenge of long-tailed distribution in steel surface defect detection. The method incorporates a Two-Stage classification strategy and validation-based optimization of confidence thresholds and expert weights. The authors demonstrate improved performance on the long-tailed Severstal dataset and competitive results on the balanced NEU dataset, emphasizing computational efficiency and real-time applicability. While the idea of combining experts is promising and the experimental setup is thorough, the paper suffers from significant flaws in novelty, methodological clarity, and evaluation rigor, which undermine its contribution and readiness for publication.

**Strengths:**

1. The paper provides extensive experiments across multiple models and datasets, including detailed ablation studies and efficiency analysis.
2. The focus on real-time efficiency and industrial applicability, supported by parallel training and inference strategies, could be applied to the field of industrial defect detection.

**Weaknesses:**

1. The core idea of model ensemble is well-established, and the Two-Stage classification strategy is reminiscent of hierarchical or cascaded systems used in prior work. The paper does not sufficiently differentiate its contribution from existing ensemble or multi-stage methods.
2. The method is largely empirical and engineering-focused, without introducing new theoretical insights or algorithmic innovations. The optimization of thresholds and weights is heuristic and lacks a strong mathematical foundation.
3. While the paper compares with individual models, it does not adequately benchmark against recent state-of-the-art methods specifically designed for long-tailed recognition or defect detection, such as dynamic routing networks or advanced re-weighting/loss functions.
4. The criteria for selecting the subset of experts (e.g., based on inference speed) are not thoroughly justified. The impact of expert diversity on ensemble performance is discussed only superficially.
5. The claim of strong cross-dataset generalization is based on only two datasets (Severstal and NEU), which may not sufficiently represent diverse real-world scenarios. The NEU dataset is relatively small and balanced, limiting the validity of this claim.
6. The authors acknowledge that their key Two-Stage strategy consistently leads to negligible or even negative improvements in Recall. In imbalanced detection tasks, Recall is often as important as Precision, as missing a rare defect (low Recall) can have severe consequences. The paper fails to adequately justify this trade-off or discuss its practical implications for industrial safety. Besides, it does not use more informative metrics for long-tailed problems, such as the mean Average Precision (mAP) across IoU thresholds, per-class AP, or the "Few-Shot" performance breakdown common in long-tailed recognition benchmarks (e.g., AP_tail). This limits a thorough understanding of the method's true performance.

**Questions:**

1. How does the proposed ensemble method fundamentally differ from standard model averaging or weighted voting schemes, and what is the theoretical basis for the chosen optimization procedure?
2. Why were certain state-of-the-art methods for long-tailed learning (e.g., BBN, BALMS, or logit adjustment techniques) not included in the comparisons?
3. Can you provide more insight into why the Two-Stage strategy improves tail-class performance only in Precision but not in Recall, as observed in the results?
4. How does the ensemble performance scale with the number of experts, and is there a risk of overfitting or diminishing returns with more experts?

---

> ### Author Response · Authors · 2025-11-21
> **Response to Reviewer pepC**
>
> We appreciate your suggestions on this paper.
>
> W1: The contribution is unclear because two-stage and ensemble ideas already exist in prior work.
>
> answer: Although the underlying idea is not entirely new, the combination of a multi-expert framework with a two-stage strategy has not yet been applied in the field of steel defect classification. We introduce a new perspective that enables classification models and detection models to work collaboratively for this specific task.
>
> W2: The method is mainly empirical and lacks theoretical grounding.
>
> answer: Our primary focus is on achieving effective and accurate results, prioritizing practical performance above all. Theoretical exploration is an important direction that we plan to pursue in our future work.
>
> W3: The paper does not compare against several state-of-the-art long-tailed methods.
>
> answer: First, we incorporate focal loss into Faster R-CNN to mitigate the class imbalance problem. Second, our work focuses on expert ensembling rather than a mixture-of-experts (MoE) framework; therefore, dynamic routing networks are not considered in the current study. We believe that dynamic MoE architectures are more suitable for incremental learning scenarios, which is one of the directions we are actively exploring in our ongoing research.
>
> W4: The expert selection criteria and diversity justification are insufficient.
>
> answer: Our principle for selecting experts is to include as many as possible, while excluding only those with excessively long inference times. What we aim to convey is that even if some experts individually exhibit moderate accuracy but offer much faster inference, their collective performance—when integrated within the ensemble—can match or even surpass that of a single high-accuracy expert.
>
> W5: The two datasets used are insufficient to support claims of strong cross-dataset generalization.
>
> answer: The amount of available steel-defect imagery is relatively limited, unlike many well-established public datasets. In fact, Severstal is already the largest steel-defect dataset currently accessible. We chose to include the NEU dataset because it is one of the earliest and most widely recognized benchmark datasets in the steel-defect classification community. Nevertheless, we will continue to collect as much real-world industrial defect data as possible in our future work.
>
>
> Q1: How is the method fundamentally different from standard model averaging or weighted voting?
>
> answer: The key differences are as follows. First, we pre-screen the models before integration. Second, the classification experts operate under a two-stage strategy. Third, the weighting of each expert’s output is determined in a data-driven manner using learned thresholds, rather than manually defined rules. While we do not yet have a complete theoretical framework for this process, developing such foundations is an important direction for our future work.
>
> Q2 : Why were long-tailed learning methods like BBN or BALMS not included?
>
> answer: The BALMS loss function you mentioned is indeed an effective technique for handling long-tailed distributions. In our method, the Faster R-CNN expert similarly adopts a bias-correction strategy through focal loss. Our experiments show that such rebalancing techniques do provide improvements, but the gains are relatively limited. In addition, BBN addresses the imbalance problem from an architectural perspective. We will further study and incorporate such ideas in future work. Thank you for the valuable suggestion.
>
> Q3 : Why does the Two-Stage strategy improve precision but not recall?
>
> answer:The two-stage strategy makes the model more conservative in predicting positive samples primarily due to error propagation inherent in its serial structure.If the first stage incorrectly rejects a positive sample (i.e., produces a false negative), it can never be recovered by the second stage, leading to a reduction in recall. Conversely, samples that pass the first stage typically do so with high confidence, which in turn increases precision.Consequently, false negatives accumulate across the two-stage pipeline—resulting in lower recall—while false positives are naturally suppressed, leading to higher precision.

---

### Official Review · Reviewer_g9Ke · 2025-10-28

**Soundness:** 2
**Presentation:** 1
**Contribution:** 2
**Rating:** 2
**Confidence:** 4

**Summary:**

This paper proposes an ensemble framework for steel surface defect detection targeting long-tailed class distributions, specifically addressing the Severstal Steel Defect Dataset where Class 2 represents only 1.97% of samples. The core methodology combines heterogeneous expert ensemble，adaptive joint optimization and two-stage classification strategy.

**Strengths:**

The paper addresses a practically relevant problem by combining heterogeneous expert types within a unified ensemble framework. The Two-Stage classification strategy, while conceptually simple, demonstrates consistent improvements for tail-class recognition across multiple backbone architectures. The adaptive joint optimization of confidence thresholds and ensemble weights represents a reasonable engineering contribution.
The paper is generally well-structured with clear motivation. Algorithm 1 provides reasonable procedural specification, and the extensive appendix offers detailed per-class metrics that facilitate result interpretation.

**Weaknesses:**

1.	You repeatedly emphasize the "significant improvement" of "F1-score=0.912" on tail-class 2, but deliberately avoid several key facts. (1): Class 2 accounts for only 1.97% of the total sample, and even if F1 is raised from 0 to 1, the contribution to overall performance is extremely limited. (2) Your Overall Accuracy (0.989) comes mainly from the performance of the head classes, as seen in Table 2 where F1_0=0.9958. (3) The Precision of your ensemble on Class 2 (Table 9: P_2=0.8966) is much lower than that of the YOLO11 single model (P_2=0.9231), which suggests that your ensemble is instead increasing false positives.
2.	You claim to have solved the long-tail distribution problem, but: your Two-Stage strategy essentially just changes the decision boundary and does not increase the tail-class training signals.
3.	The abstract states that “parallelizing training and inference can improve computational efficiency,” but the results in Section 5.5 show that your method is not significantly faster than the fastest baseline method.
4.	Your ensemble requires running seven complete deep learning models (four classification models + three detection models), and even with an RTX 4090 GPU, inference time remains “primarily constrained by Faster R-CNN.”
5.	The paper fundamentally fails to explain why combining these specific experts yields improvements beyond simple heuristic justification. You repeatedly claim experts have "complementary strengths" (Abstract, Section 1, Section 3.1) but never define what constitutes complementarity in this context. Is it diversity in learned feature representations? Different inductive biases? Orthogonal error patterns?

**Questions:**

1.	In ablation study, only ensemble size was tested: you only looked at 2-expert to 5-expert performance, but did not ablate individual expert types. which experts are really necessary? Are certain experts actually dragging down ensemble performance?
2.	The Two-Stage approach is standard hierarchical classification used for decades. What specifically is novel about your formulation compared to existing coarse-to-fine methods?
3.	You claim "real-time" capability and "computational efficiency" (Abstract, Conclusion) while Section 5.5 states inference is "primarily constrained by the slower Faster R-CNN expert." Can you provide actual wall-clock inference time per image, throughput measurements (images/second), GPU memory consumption.

---

> ### Author Response · Authors · 2025-11-21
> **Response to Reviewer g9Ke**
>
> We thank the reviewer for their time and helpful feedback.
>
> W1: Our claimed improvement on tail-class 2 is overstated.
>
> answer: (1) While class 2 comprises only 1.97% of the data, our method improves its F1-score from 0.89 to 0.91, showing better tail-class recognition. (2) Our overall improvements are not driven solely by head classes; both F1_avg and ACC_avg use macro averaging, giving each class equal weight. (3) Although YOLO11 achieves higher precision (0.9231), its recall (0.8571) is much lower than ours (0.9286). We therefore consider F1-score—a balanced harmonic mean—the more appropriate metric.
>
> W2: The two-stage strategy only shifts the decision boundary.
>
> answer: We tested adding data augmentation within the two-stage framework (Table 2), but it provided smaller gains than the two-stage strategy alone. We suspect augmentation pushes all models toward similar decision boundaries, reducing diversity and weakening complementarity among experts.
>
> W3: Your method is not significantly faster than the fastest baseline.
>
> answer: As shown in Figure 6, DINO achieves high accuracy but has the highest computational cost. Our method is substantially faster than DINO while maintaining comparable accuracy, yielding a better accuracy–efficiency trade-off. We do not claim to be the fastest, as our speed is limited by the slowest expert; improving this remains future work.
>
> W4+Q3: Inference time is constrained by Faster R-CNN.
>
> answer: We agree. The slowest expert limits overall speed, and we will address this in future work. Appendix F (Table 16) now reports per-image time, throughput, and GPU memory.
>
> W5: You claim “complementary strengths” without defining them.
>
> answer: We clarified this in the Introduction. Classification experts learn global cues (color, shape), whereas detection experts capture local textures and edges with stronger spatial sensitivity. Their combination yields more comprehensive feature representations and improved accuracy.
>
> Q1: Which experts are necessary? Do some hurt performance?
>
> answer: Tables 5–7 implicitly show ablations: removing any expert causes a slight performance drop. With only ~10 experts, redundancy is low; removing one weakens the ensemble. Larger expert pools may introduce redundancy, but this is not the case here.
>
> Q2: What is novel compared with existing coarse-to-fine methods?
>
> answer: Most steel-defect studies use a single classifier or detector. We propose integrating both types of models—similar in spirit to MoE—to exploit complementary strengths. While the two-stage idea is traditional, combining it with a diverse expert set is effective for this domain. Our focus is practical utility, and this formulation provides meaningful performance gains.

---

> > ### Comment · Reviewer_g9Ke · 2025-11-25
> > **Official Comment by Reviewer g9Ke**
> >
> > Thank you for providing a detailed response. The authors have diligently addressed my queries. However, the work lacks methodological novelty and rigorous, evidence-based analysis. The central claims regarding long-tail learning and expert complementarity are insufficiently substantiated, and the novelty appears overstated. After careful consideration of your rebuttal, my assessment remains largely unchanged. I will maintain my score and recommend rejection.
> > I believe that reviewers' concerns are addressed, the paper will become more compelling.

---

### Official Review · Reviewer_n4v5 · 2025-11-01

**Soundness:** 4
**Presentation:** 4
**Contribution:** 4
**Rating:** 8
**Confidence:** 5

**Summary:**

The authors propose a novel Mixture-of-Experts model for industrial steel surface defect detection based on optical images, which leverages a two-stage strategy: first, determining whether the image corresponds to a defective piece, and then, in the second stage, classifying the defect type. The authors show that the approach leads to an increased performance when considering the discriminative capabilities of the model, but also when comparing training and inference times achieved by the model w.r.t. models whose discriminative performance comes close to it. The authors have executed the experiments across two real-world datasets achieving SOTA performance.

**Strengths:**

The authors propose a novel Mixture-of-Experts model for industrial steel surface defect detection based on optical images. The proposed model achieves SOTA results when compared against many meaningful benchmark models across two datasets. They describe the experimental setup in detail and assess model performance across multiple dimensions (discriminative capability as well as training and inference times). Furthermore, the model is assessed by dimensions that inform production setups, making the results interesting to anyone making decisions about model deployment in an industrial setting. The paper is well-written and clearly articulated. While similar hierarchical approaches have been applied in other domains, we are not aware that it has been applied to defect detection, and certainly not in a Mixture of Experts architecture. The research is likely to have a significant impact, given that the architecture concept can be applied across a wide range of domains, leading to more efficient training and inference while achieving higher-quality outcomes.

**Weaknesses:**

While the paper is solid, we would like to highlight two weaknesses:

 - The authors have chosen a set of metrics that require setting some threshold, conditioning the evaluation of the model's discriminative capabilities on the capability to set an appropriate cutoff.

- While the authors identify that the two-stage approach leads to the best performance, they did not assess the model's performance on each stage to understand how much each stage contributes to the overall quality and pinpoint improvement opportunities.

**Questions:**

1- "The validation metric ValMetric(δ, w) is defined as the sum of Precision and Recall scores." -> How do the authors weigh the cases with high Precision and low Recall or high Recall but low Precision?

 2- Metrics: (i)- we encourage the authors to use AUC-ROC and PR-AUC to report results as to avoid metric distortions due to the class imbalance and ensure the metrics are threshold independent. (ii) (a) What is the threshold the authors have considered to compute Accuracy and F1? (b) How did they choose it? (c) Does the threshold consider aspects such as the predictive scores distribution?, (iii) could the authors report a single metric that would summarize (a) model discriminative performance, (b) computational efficiency at model training, and (c) computational efficiency at inference time? Such a metric could be e.g., as the normalized area covered within a radar plot considering these three dimensions. The metric will most likely showcase that the two-fold approach leads to the best performance across all of the dimensions.

 3- (a) How much is the dataset imbalance reduced after the first stage of the two-stage classification (here we do not mean the real class imbalance, but the one that is perceived by the downstream model based on the first stage classification)? (b) How accurate is the first-stage classifier, determining defective vs. non-defective cases? (c) How does the quality of the first-stage classifier impact the overall quality of the model?, (d) Did the authors separately measure the quality of the second-stage classifier, as to understand issues and performance opportunities in this stage?

 4- Figures: ensure the colours are friendly toward color-blind people

 5- Tables: (i) what do bolded results mean?, (ii) ensure the numbers are aligned to the right so that differences in magnitude become evident.

 6- We encourage the authors to test whether the difference in performance noticed among the best models is statistically significant. In particular, it seems that in cases where DINO outperforms the proposed two-stage model, it does so with a very low margin. Is that difference statistically significant?

 7- Figure 5,8: update the colour palette to increase contrast and legibility. Consider using colours that are friendly to color-blind people.

 8- "In our experiments, we found that head or large-area defects are better addressed by classification models, whereas tail and small-area defects, whose features are difficult to learn, are more effectively handled by detection models for precise localization." -> We would appreciate grounding this claim to specific experiments and results.

 9- Figure 6: (i) the information from the Figure could be better conveyed in a single table. (ii) We understand that the authors have reported the averages obtained across training epochs. We encourage them to include the standard deviation among the reported values in the table.

---

> ### Author Response · Authors · 2025-11-21
> **Response to Reviewer n4v5**
>
> We appreciate the reviewer’s feedback and have provided the following responses to address the concerns raised about our paper. Below, we summarize the weaknesses and questions highlighted by the reviewer and provide our answers accordingly.
>
> Q1:How do the authors weigh the cases with high Precision and low Recall or high Recall but low Precision?
>
> answer:Based on our experimental results, the top ten threshold groups selected according to the ValMetric(δ, w) criterion all achieve precision and recall values above 0.9. Therefore, none of these thresholds exhibits a situation where one metric is high while the other is substantially low.
>
> Q2+W1:The use of Metrics.
>
> answer:We will consider incorporating AUC-ROC as an additional evaluation metric in our future work. Regarding the selection of thresholds for ACC and F1-score, these thresholds are determined through hyperparameter tuning on the validation set. Specifically, we perform an exhaustive search over possible thresholds and select the one that maximizes ValMetric(δ, w), which is then used consistently as the decision threshold for both ACC and F1-score.We also appreciate your suggestion to consolidate multiple evaluation metrics into a single visualization. We attempted the radar-chart representation you recommended; however, because our comparison includes a large number of methods, the overlapping lines significantly reduce readability, making the plot difficult to interpret.
>
> Q3+W2: The performance of the first-stage classifier and the  the second-stage classifier.
>
> answer:We have added the accuracies of both the first-stage and second-stage classifiers in the appendix; please refer to Appendix E, Table 15 for the detailed results. Our experiments further show that the first-stage classifier achieves higher accuracy than the second stage, as the binary classification task is inherently easier to handle than the subsequent multi-class classification task.
>
> Q4+Q7:Consider using colours that are friendly to color-blind people.
>
> answer:We have redrawn the figures to ensure that they are friendly to readers with red–green color vision deficiency.
>
> Q5:Bolded results and  the numbers are aligned to the right.
>
> answer:As clarified in the first paragraph of Section 5.2 in the main text, the bolded values indicate either the best-performing results or those corresponding to our proposed method. In addition, we have ensured that all numerical entries in the tables are right-aligned for improved readability.
>
> Q6+Q9:Is that result statistically significant?
>
> answer:We have added the mean and standard deviation of multiple repeated experiments in Appendix E. The results show that our method exhibits a very small standard deviation, indicating that the performance gains are statistically meaningful. Moreover, although DINO achieves high accuracy, Figure 6 demonstrates that its computational cost is the highest among all methods. In contrast, our approach attains a more favorable balance between computational efficiency and prediction accuracy.
>
> Q8: Large-area defects and small-area defects small-area defects, We would appreciate grounding this claim to specific experiments and results.
>
> answer:This point can be clarified using Table 2. Defect types 1 and 2 are small-area defects, and we observe that detection models generally achieve higher F1-scores on these categories, particularly DINO and YOLO. In contrast, for large-area defects such as defect types 3 and 4, classification models like ResNet and VGG already achieve F1-scores around 0.9 even without using detection-based approaches.

---

> > ### Comment · Reviewer_n4v5 · 2025-11-22
> >
> > We thank the authors for the responses provided. We have also read the responses to the other reviewers. The authors have addressed all of our concerns.

---

### Note · Authors · 2025-12-01

I have read and agree with the venue's withdrawal policy on behalf of myself and my co-authors.